# Sea ice classification of TerraSAR-X ScanSAR images for the MOSAiC expedition incorporating per-class incidence angle dependency of image texture

Wenkai Guo[1], Polona Itkin[1], Suman Singha[2], Anthony Paul Doulgeris[1], Malin Johansson[1], and Gunnar Spreen[3]

[1]Department of Physics and Technology, UiT The Arctic University of Norway
[2]Remote Sensing Technology Institute (IMF), German Aerospace Center (DLR), 28359 Bremen, Germany *Currently with National Center for Climate Research (NCKF), Danish Meteorological Institute (DMI), 2100 Copenhagen, Denmark
[3]Institute of Environmental Physics, University of Bremen

**Correspondence:** Wenkai Guo (wenkai.guo@uit.no)

**Abstract.**

We provide sea ice classification maps of a sub-weekly time series of single (HH) polarization X-band TerraSAR-X ScanSAR (TSX SC) images from November 2019 to March 2020 covering the Multidisciplinary drifting Observatory for the Study of Arctic Climate (MOSAiC) expedition. This classified time series benefits from the wide spatial coverage and relatively high spatial resolution of TSX SC data and is a useful basic dataset for future MOSAiC studies on physical sea ice processes and ocean and climate modeling. Sea ice is classified into leads, young ice with different backscatter intensities, and first-year ice (FYI) or multi-year ice (MYI) with different degrees of deformation. We establish per-class incidence angle (IA) dependencies of TSX SC intensities and Gray-Level Co-occurrence Matrix (GLCM) textures, and use a classifier that corrects for the class-specific decreasing backscatter with increasing IAs, with both HH intensities and textures as input features. Optimal parameters for texture calculation are derived to achieve good class separation while keeping maximum spatial detail and minimizing textural collinearity. Class probabilities yielded by the classifier are adjusted by Markov Random Field contextual smoothing to produce classification results. The texture-based classification process yields an average overall accuracy of 83.70% and good correspondence to geometric ice surface roughness derived from *in situ* ice thickness measurements (correspondence consistently close to or higher than 80%). A logarithmic positive relationship is found between geometric ice surface roughness and TSX SC HH backscatter intensity, similar to previous C- and L-band studies. Areal fractions of classes representing ice openings (leads and young ice) show prominent increases in mid-late November 2019 and March 2020, corresponding well with ice opening time series derived from *in situ* data in this study and those derived from satellite SAR and optical data in other MOSAiC studies.

## 1 Introduction

During the one-year-long Multidisciplinary drifting Observatory for the Study of Arctic Climate (MOSAiC) expedition from 2019 to 2020, the icebreaker RV Polarstern drifted with sea ice along the Transpolar Drift in the Central Arctic, conducting the

largest multidisciplinary Arctic research expedition in history (Nicolaus et al., 2022). Satellite data acquisitions from multiple platforms were coordinated to survey the sea ice area surrounding the expedition, enabling continuous large-scale sea ice monitoring along the drift. Also, extensive on-ice, airborne and ship-based *in situ* data were collected surrounding the MOSAiC ice floe, where Polarstern was moored and the Central Observatory (CO) established. These include data from meteorological stations, airborne laser surveys, ship radar measurements, and a distributed network of autonomous buoys, etc (Krumpen and Sokolov, 2020; Nicolaus et al., 2021; Shupe et al., 2022). This expedition aimed to facilitate physical, biogeochemical and ecological studies of the region, enabling multi-scale quantification of relevant processes and feedbacks and eventually the production of improved climate and Earth system models (Krumpen et al., 2021; Nicolaus et al., 2021; Shupe et al., 2022).

Sea ice type classification is an important basic representation of sea ice conditions which supports various further analyses, e.g., monitoring ice break-up and lead formation, inferring the occurrence of sea ice deformation, studying ice-associated and under-ice ecology, and as input to ocean and climate models, etc. Satellite Synthetic Aperture Radar (SAR) data has been used widely for sea ice classification for operational and scientific purposes due to its high spatial resolution and weather- and illumination-independent monitoring capabilities (Zakhvatkina et al., 2019). Coordinated acquisitions of TerraSAR-X ScanSAR (TSX SC) data were conducted to provide consistent coverage of the MOSAiC ice floe throughout the expedition. This dataset provides daily X-band (9.65 GHz) imaging with 8.25 m nominal pixel spacing, which is considerably higher than publicly available ScanSAR products, e.g., Sentinel-1 (S1). The extents of TSX SC scenes are approximately $100 \times 150$ km. These features make TSX SC a valuable data source for detailed examinations of sea ice development for large areas around MOSAiC. This study aims to produce a classified winter (November 2019 to March 2020) TSX SC time series surrounding the CO, which can serve as a basis for further MOSAiC sea ice studies and modeling efforts.

TSX SC scenes in this time series cover a wide range of incidence angles (IAs). Therefore, appropriate adjustment for the IA effect of SAR signal, i.e., generally decreasing backscatter intensities with IA, is crucial for reliable and consistent classification. The magnitude of the IA effect varies with ice types (Mäkynen et al., 2002; Mäkynen and Karvonen, 2017; Mahmud et al., 2018), which necessitates per-class IA correction. A sea ice classifier which specifically considers this phenomenon is used in this study. Developed and published by Lohse et al. (2020), this classifier directly incorporates per-class IA dependencies into a Bayesian classifier, treating the IA dependence as a class property. This classifier replaces the constant mean vector of the Gaussian probability density function with a linearly variable mean, which represents class-specific IA dependencies (Lohse et al., 2020), and is therefore named the Gaussian Incidence Angle (GIA) classifier. It has been developed based on S1 ExtraWide (EW) data and has also been used with Radarsat-2 ScanSAR Wide and Fine Quad-pol (RS2 SCW and FQ) data with minor adjustments (Guo et al., 2022). The GIA classifier reliably corrects the IA effect on HH and HV channels of these datasets, resulting in improved classification results compared to classification with global IA correction.

TSX SC data collected for MOSAiC is in HH polarization. The same ice types can have vastly different HH intensities due to different surface characteristics, e.g., different degrees of deformation on first-year ice (FYI) and multi-year ice (MYI), and different surface roughness and salinity levels on young ice, etc. On the other hand, some ice types have similar X-band HH intensities, e.g., MYI, deformed FYI, and young ice (e.g., Liu et al. 2016). Therefore, in addition to HH intensities, we use image textures as input to the classification to expand the feature space. Specifically, we use texture measures calculated on the

basis of the gray-level co-occurrence matrix (GLCM, Haralick et al. 1973). The GLCM tabulates how different combinations of gray-levels co-occur in image windows, based on which statistical measures are derived to represent spatial variability surrounding the central pixel. GLCM textures are among the most powerful texture discrimination tools (Barber and LeDrew, 1991; Zakhvatkina et al., 2019) and have been widely used for texture-based classification of remote sensing images in general (Hall-Beyer, 2017) and specifically for sea ice classification of X- and C-band SAR data (e.g., Clausi and Yu 2004; Leigh et al. 2014; Zakhvatkina et al. 2017; Murashkin et al. 2018; Park et al. 2020; Lohse et al. 2021, and those listed in Table 1). Compared to classification based only on SAR intensities, they provide additional separability between FYI and MYI, young ice and MYI, and level and deformed ice (e.g., Holmes et al. 1984; Shokr 1991; Leigh et al. 2014; Zakhvatkina et al. 2017; Lohse et al. 2021).

Table 1 shows GLCM texture parameters used in previous sea ice classification studies using X-band SAR. Texture names and parameters can be found in Haralick et al. (1973) and Conners and Harlow (1980), and are introduced in more details in Section 2.3. The table shows a wide variety of datasets and parameters, indicating that various GLCM textures on different geographical scales are useful for discriminating between sea ice classes. Many studies use a limited number of textures measures and do not involve a process of selecting texture combinations based on class separability and textural collinearity.

**Table 1.** Texture parameter selection in previous studies of X-band SAR sea ice classification.

| | Data | | | | Texture parameters | | |
|---|---|---|---|---|---|---|---|
| | **Area** | **Dataset** | **Frequency & channel**[1] | **Resolution**[2] **(m)** | **GLCM textures**[3] | **Window size**[3] **- pixel (m)** | **Co-occurrence distance**[3] **- pixel (m)** |
| Holmes et al. (1984) | Beaufort Sea | SURSAT SAR-580 (airborne) | X-band HV | 3 | CON, ENP | 5 (15) | 2 (6) |
| Barder & LeDrew (1991) | Mould Bay, Canada | STAR-1 (airborne) | X-band HH | 6 | UNI[4], COR, ENP, DIS, CON | 25 (150) | 1 (6) |
| Shokr (1991) | Mould Bay, Canada | STAR-1 (airborne) | X-band HH | 36 | CON, ENP, UNI[4], HOM, MAX | 5 (180) | 1 (36) |
| Liu et al. (2016) | East coast, Antarctica | TSX SC/WSC | X-band HH | 15 | ASM, CON, COR, DIS, ENP, HOM, MEAN, VAR | 39 (585) | 4 (60) |
| Ressel et al. (2015) | Barents Sea | TSX SC | X-band VV | ~48 | CON, DIS, ENG, ENP, HOM | 11 (~528) | 1 ( 48) |
| Zhang et al. (2019) | Barents Sea | TSX SC | X-band HH/VV | 8.25 | CON, COR, HOM, MEAN, VAR | 39 (321.75) | 4 (33) |
| Liu et al. (2021) | Beaufort Sea | TSX SC/WSC | X-band HH | 8.25 | CON, COR, HOM, MEAN, VAR | 39 (321.75) | 4 (33) |

[1] Only SAR channels used for GLCM calculation are shown.
[2] Effective pixel spacing after pre-processing.
[3] GLCM textures, window sizes, and co-occurrence distances are those used for texture-based classification, or those that yield best classification results in studies comparing different parameter combinations.
[4] UNI: Uniformity = $\sum_i \sum_j P_{i,j}^2$, therefore similar to ENG.

In the logarithmic (dB) domain, S1 EW textures of the HH channel for different ice types generally have a linear relationship with IA, and have been used for sea ice classification (Lohse et al., 2021). For TSX data, Ressel et al. (2015) used 5 GLCM textures from the VV channel of 3 TSX SC images to classify sea ice near Svalbard with an artificial neural network (ANN)

and reported satisfactory results for scenes with similar IA ranges to the training scene. Liu et al. (2016) used 8 GLCM textures from TSX SC and Wide ScanSAR (WSC) data to classify sea ice on the east coast of Antarctica, using IA directly as an input feature to a support vector machine (SVM) classifier. Zhang et al. (2019) used 5 GLCM textures in an SVM classifier on 5 TSX SC (HH/VV) scenes, and Liu et al. (2021) used the same 5 GLCM textures on 8 TSX SC/WSC (HH) scenes to classify sea ice, both in the Beaufort Sea, with no corrections for the IA effect. To our knowledge, no previous study has demonstrated IA dependencies of different Arctic sea ice types for TSX SC intensities and GLCM textures.

This study examines this phenomenon in winter MOSAiC and accordingly includes GLCM textures as input features to the GIA classifier. Optimal parameters for texture calculation are derived to provide statistical separability between class distributions evaluated by the Kolmogorov-Smirnov (K-S) distance (Massey Jr, 1951). 17 GLCM texture measures are analyzed, which are derivable using commonly available software tools, i.e., ESA SNAP (European Space Agency, 2020) and the Google Earth Engine (GEE, Gorelick et al. 2017). As we aim to fully utilize the spatial resolution of TSX SC data, a rating system is developed to find the set of texture measures that provides class separability at the smallest window size while minimizing inter-correlations between textures.

In summary, the objectives of this study are: 1. to use the GIA classifier on TSX SC HH intensity and textures to produce a classified winter time series for sea ice surrounding the MOSAiC expedition; 2. to demonstrate per-class IA dependencies of TSX SC HH intensity and textures for the above mentioned study area and period.

## 2 Materials and methods

### 2.1 Data

This study analyzes 53 TSX SC scenes (2019.11.01 to 2020.04.11, IA: 17.18° to 59.56°) with an average of 3 scenes per week. All scenes are radiometrically corrected and calibrated to $\sigma^0$ and converted to dB. Fig. 1(a) shows the scene boundaries in each month as black rectangles. Fig. 1(c) shows IA ranges of the scenes in red.

Among these, 50 scenes (2019.11.01 to 2020.03.28, IA: 31.90° to 59.56°) are used for sea ice classification, and are hereafter referred to as the time series. The remaining 3 scenes (2020.03.31, 2020.04.03 and 2020.04.11, IA: 17.18° to 36.70°) are only included to cover the full IA range of TSX SC data in examining IA dependencies of HH intensities. They were captured at low IAs (Fig. 1(c)) to keep the CO, which was drifting below 85.5°N (Fig. 1(b)), in the scene frames. These scenes exhibit consistent IA dependency with other scenes for HH intensities but not for HH textures (not shown). The spatial details obtainable from these scenes are different from others after being subjected to identical pre-processing steps, resulting in considerably different texture values. Also, these scenes are generally more affected by noise (Fritz et al., 2013).

We use 13 scenes, including these 3 scenes with low IAs, as reference scenes (dates and IA ranges shown in Fig. 1(c) and (d)), from which reference polygons are derived to examine IA dependencies. These scenes are selected to cover each month between November 2019 and April 2020 and the full IA range of TSX SC data. As mentioned above, the 3 scenes with low

IAs are used only for demonstrating IA dependencies of HH intensities, and the other 10 reference scenes before 2020.03.31 are used for demonstrating IA dependencies of both HH intensities and textures, and for classification training and testing,

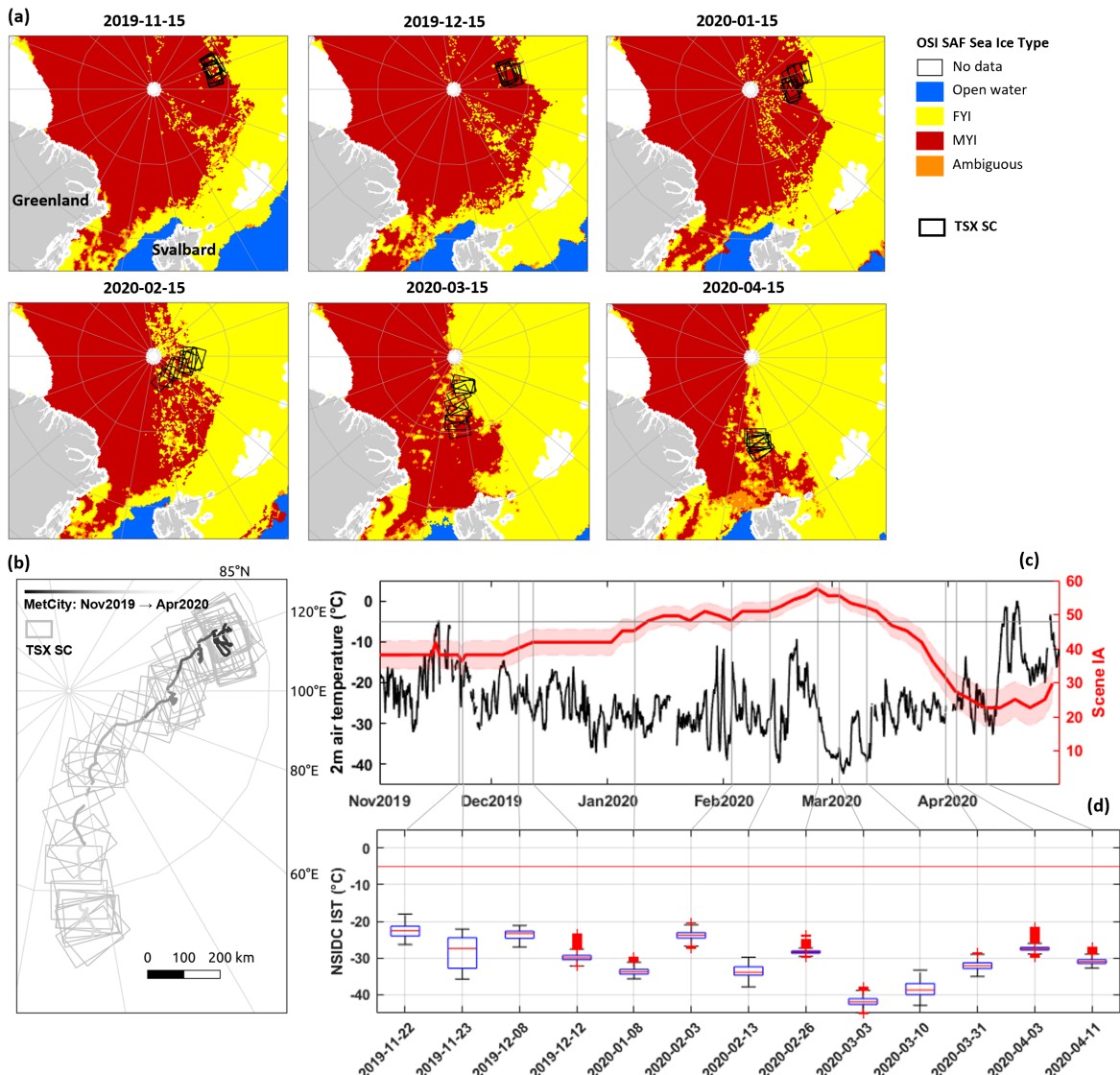

**Figure 1.** (a) TSX SC scenes in each month and OSI SAF sea ice classifications of surrounding sea ice areas in the middle of each month; (b) the drift track of the weather station MetCity and its relative position to TSX SC scenes; (c) 2m air temperature records and IA ranges of TSX scenes (average IAs as the red line), with vertical lines representing selected reference scenes; (d) box-plots of NSIDC IST within each reference scene, where boxes cover the $25^{th}$ to the $75^{th}$ percentile with the median shown as the red bar. Whiskers extend to data extremes excluding outliers, and red crosses indicate outliers.

Environmental conditions associated with the scenes are inferred from 2m air temperature records from the weather station MetCity in the CO (Fig. 1(c), black), the drift track of which is shown in Fig. 1(b) as the gray line. Fig. 1(c) shows that temperatures are mostly below $-5°C$ through the study period except for late April, when warm spells brought temperatures to near $0°C$. For the reference scenes, near-coincident scenes (within 3 hours from TSX acquisition) from the National Snow and Ice Data Center (NSIDC) MOD29/MYD29 sea ice surface temperatures (IST) dataset (Hall and Riggs., 2021) are extracted to show that temperatures within TSX scene boundaries are well below -5°(Fig. 1(d)). Overlapping S1 EW and RS2 FQ scenes and the Ocean and Sea Ice Satellite Application Facility (OSI SAF) sea ice type (OSI-403-d, Fig. 1(a)) product (OSI SAF, 2019) are used as qualitative visual reference to aid the derivation of reference polygons, providing general knowledge about large-scale ice conditions and comparison with C-band SAR signals, respectively.

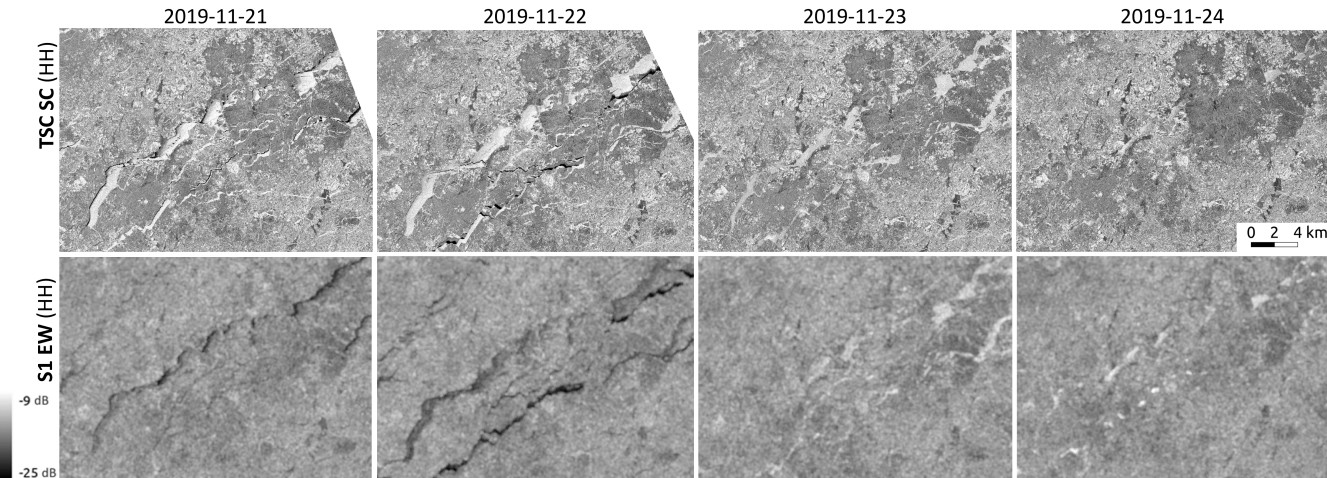

**Figure 2.** Progression of young ice on near-coincident TSX SC and S1 EW scenes scaled by the same range of intensities. All subsequent figures of HH intensities use the dB range shown here.

Young ice shows a wide range of HH intensities due to differences in surface characteristics, which affects ice type classification. Fig. 2 shows an example of the progression of young ice on overlapping TSX SC and S1 EW scenes in HH polarization. On 2019.11.20, wide-spread lead openings occurred around the CO. Between 2019.11.20 and 2019.11.21, more openings appeared which quickly re-froze into young ice. On the TSX scenes, on 2019.11.21, most of the young ice areas appear very bright. Subsequently, young ice gradually darken to similar brightness to the surrounding ice. On the S1 scenes, HH intensities of young ice gradually increase, from similar or lower brightness to nearby level ice to very bright on 2019.11.23 and 2019.11.24. Afterwards, they again darken to similar brightness to the surroundings. The changing young ice intensities are presumably due to evolving surface roughness, e.g., influenced by the formation and evolution of frost flowers which are highly saline with varying sizes, leading to varying scales of surface roughness (Martin et al., 1995; Barber et al., 2014; Isleifson et al., 2018; Johansson et al., 2018). The delayed increase and decrease in young ice backscatter in C-band (5.405 GHz) compared to X-band (9.65 GHz) is then presumably due to different interactions between changing surface roughness scales and different

SAR wavelengths (Isleifson et al., 2010; Dierking, 2010; Barber et al., 2014; Park et al., 2020). These observations confirm the need to split young ice into separate classes for ice type classification, which is described below.

## 2.2 Reference polygons of sea ice classes

Based on the ice conditions in the study area and period, we classify sea ice into leads, rough young ice with different HH intensities, and FYI or MYI with different deformation states. Intensity thresholds shown below are empirically derived approximate values only used as one of the criteria in deriving the reference polygons.

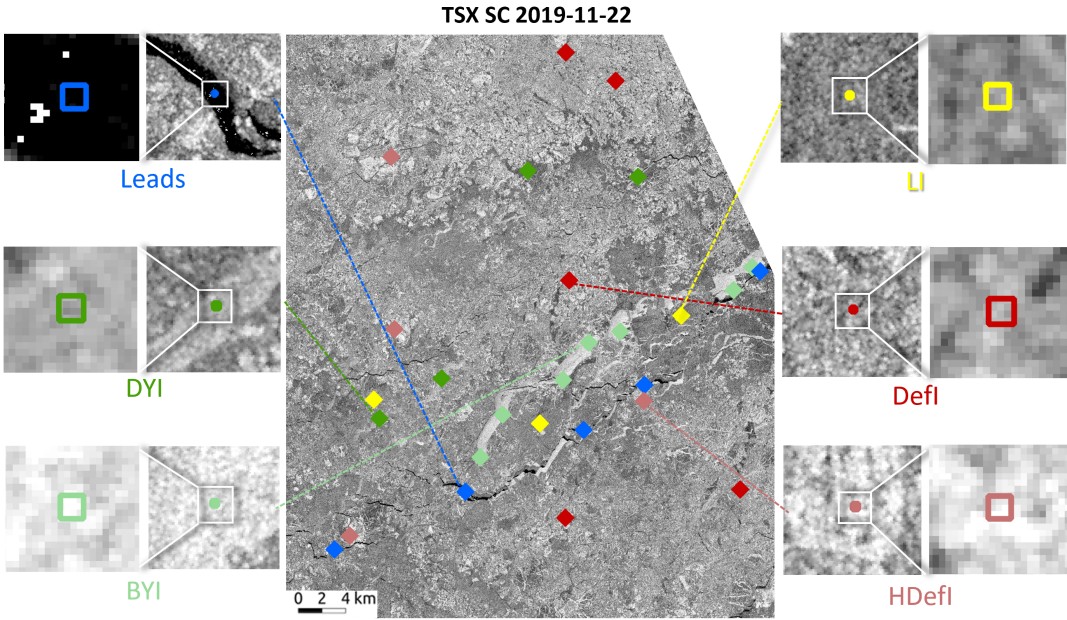

**Figure 3.** Example reference polygons of different classes over the scene on 2019.11.22.

    1. Leads: ice openings occupied by calm open water, nilas or smooth newly formed ice, having the lowest HH intensities ($< -25$ dB). The separation between open water in different wind states is not within the scope of this study, and a visual

examination shows that open water leads in the time series are all narrow ($\leq 250\,m$) and predominantly in a calm state.

    2. Dark young ice (DYI): newly formed ice in leads with relatively high HH intensities ($\geq -15$ dB) are all regarded as young ice, irrespective of thickness. Young ice is further split into two classes, as mentioned above, with the DYI class having comparatively low intensities (between $-15$ dB and $-10$ dB). The separated young ice classes do not correspond to existing ice types in the WMO nomenclature (WMO, 2018).

3. Bright young ice (BYI): rough young ice with HH intensities of greater than $-10$ dB.

    4. Level ice (LI): smooth FYI or MYI areas having intermediate HH intensities, between leads and DYI ($-25$ dB and $-15$ dB).

5. Deformed ice (DefI): rough FYI or MYI with HH intensities between $-15$ dB and $-10$ dB.

6. Heavily deformed ice (HDefI): FYI or MYI areas with very high degrees of deformation, thus having high HH intensities ($\geq -10$ dB).

For each class, 15 reference polygons in rectangles of $3{\times}3$ pixels are manually derived for each reference scene to standardize the number of reference pixels between classes. The polygon size is determined to accommodate typical widths of small or linear surface features, i.e., classes representing 'lead ice' (leads, DYI and BYI) and also HDefI. The former usually takes a linear shape along ice openings, and the latter usually includes linear strips or spatially limited aggregations of deformation features, or rounded MYI floes. Therefore, polygons are placed at the center of small, rounded features and along the width of linear features. To minimize spatial dependence, a minimum distance of 50 pixels is kept between polygons, and polygons for each class are distributed evenly across the scenes where possible. Polygons of each class in each scene are then randomly split in half for training and testing. To improve training consistency across scenes, polygons of LI, DefI and HDefI are derived for approximately the 'same ice' for all reference scenes, where possible. Fig. 3 shows example reference polygons derived for 2019.11.22.

## 2.3 IA dependencies of HH intensities and textures

We examine IA dependencies of HH intensities and 17 GLCM textures for different ice types and evaluate class separability provided by them. This enables us to optimize the utilization of GLCM textures as classification features and classify sea ice for MOSAiC with reliable IA correction. The textures used are listed in Table 2, where the mathematical expressions match those from Haralick et al. (1973) and Conners and Harlow (1980).

In an initial examination of GLCM textures, we found that only textures of HH intensities in the logarithmic (dB) domain have a consistent linear relationship with IA, given properly constrained IA range (more details below), while textures of HH intensities in the linear domain do not. This linear dependency is one of the pre-requisites for input features of the GIA classifier. Similar findings are reported in Lohse et al. (2021) for C-band S1 EW data. Thus, GLCM textures are calculated for HH intensities in dB, split into 64 gray levels to achieve balance between precision of gray-level information and computational efficiency, averaged for 4 directions ($0°$, $45°$, $90°$, $135°$) to avoids directional sensitivity of textures. We use a data-driven approach to optimize the other 3 texture parameters for image classification: co-occurrence distance, texture window size and the combination of texture measures (more details shown in Section 2.4). These texture parameters are explained in Haralick et al. (1973).

Statistical distribution and scatter plots of HH intensities of the 13 reference scenes (Fig. 4) show that ambiguities in HH intensities are most prominent for two class pairs: BYI vs. HDefI and DYI vs. DefI. These difficult class pairs are thus the focus of subsequent separability evaluations. The IA dependency of leads is the weakest and statistically insignificant (Fig. 4), with HH intensities mostly under the nominal noise floor (Fritz et al., 2013) and having the widest scatter. HH intensities of other classes are generally linear with IA through the whole IA range with significant slopes.

The distribution of GLCM textures in an example window size of 9 pixels and their scatter plots in the IA range of the 10 training scenes ($31.90°$ to $59.56°$) are shown in Fig. 4. Only the difficult class pairs are shown for visual clarity. Textures

**Table 2.** GLCM texture measures analyzed in this study, where $P_{i,j}$ is the $(i,j)$th entry in the GLCM; $\sum_i$ is $\sum_{i=1}^{N_g}$; $\sum_j$ is $\sum_{j=1}^{N_g}$; $N_g$ is the number of distinct gray levels in the quantised image; $P_{x+y}(k) = \sum_{i=1}^{N_g} \sum_{j=1}^{N_g} P_{i,j}$ $(i+j=k; k=2,3,\ldots,2N_g)$; $P_{x-y}(k) = \sum_{i=1}^{N_g} \sum_{j=1}^{N_g} P_{i,j}$ $(|i-j|=k; k=0,1,\ldots,N_g-1)$; $P_x(i) = \sum_{j=1}^{N_g} P_{i,j}$; $P_y(j) = \sum_{i=1}^{N_g} P_{i,j}$; $\mu_x$, $\mu_y$, $\sigma_x$ and $\sigma_y$ are the means and standard deviations of $P_x$ and $P_y$; $HXY = -\sum_i \sum_j P_{i,j} log P_{i,j}$; $HXY1 = -\sum_i \sum_j P_{i,j} log p_x(i) p_y(j)$; $HXY2 = -\sum_i \sum_j p_x(i) p_y(j) log p_x(i) p_y(j)$; and $HX$ and $HY$ are entropy of $P_x$ and $P_y$.

---

(1) Cluster Prominence (CLP):

$\sum_i \sum_j (i+j-\mu_i-\mu_j)^4 P_{i,j}$

(2) Cluster Shade (CLS):

$\sum_i \sum_j (i+j-\mu_i-\mu_j)^3 P_{i,j}$

(3) Contrast (CON): $\sum_i \sum_j P_{i,j}(i-j)^2$

(4) Correlation (COR): $\frac{\sum_i \sum_j ij P_{i,j} - \mu_x \mu_y}{\sigma_x \sigma_y}$

(5) Difference Entropy (DFE):

$-\sum_{i=0}^{N_g-1} P_{x-y}(i) log P_{x-y}(i)$

(6) Difference Variance (DFV):

$\sum_{i=2}^{2N_g} \left( i - \left[ \sum_{i=2}^{2N_g} i P_{x-y}(i) \right] \right)^2$

(7) Dissimilarity (DIS): $\sum_i \sum_j P_{i,j} |i-j|$

(8) Energy (ENG): $\sqrt{\sum_i \sum_j P_{i,j}^2}$

(9) Entropy (ENP): $\sum_i \sum_j P_{i,j} (-ln P_{i,j})$

(10) Homogeneity (HOM): $\sum_i \sum_j \frac{P_{i,j}}{1+(i-j)^2}$

(11) Information Measure of Correlation 1 (IMC1):

$\frac{HXY - HXY1}{max(HX,HY)}$

(12) Information Measure of Correlation 2 (IMC2):

$\sqrt{(1 - exp(-2.0(HXY2-HXY)))}$

(13) Maximum Probability (MXP): $max(P_{i,j})$

(14) Mean (MEAN): $\sum_i \sum_j i P_{i,j}$

(15) Sum Average (SMA): $\sum_{i=2}^{2N_g} i P_{x+y}(i)$

(16) Sum Variance (SMV):

$\sum_{i=2}^{2N_g} \left( i - \left[ \sum_{i=2}^{2N_g} i P_{x+y}(i) \right] \right)^2$

(17) Sum of Square: Variance (VAR):

$\sum_i \sum_j P_{i,j} (i-\mu)^2$

---

generally show a weak linear relationship with IA with varying levels of dependencies (IA slopes), similar to previous C-band and X-band findings (e.g., Liu et al. 2016; Lohse et al. 2021; Scharien and Nasonova 2020). Some textures show visually apparent separability between one or both of the difficult class pairs (e.g., DIS, ENP, MEAN, SMA, VAR). The classes form approximately Gaussian distributions for HH intensities and most textures (Fig. 4), satisfying the pre-requisite for input features of the GIA classifier.

A considerable part of the leads class is below the nominal noise floor, affecting its distribution for HH intensities and textures. Also, the leads class has distinctly different HH intensities than other classes. Therefore, leads is excluded from sub-

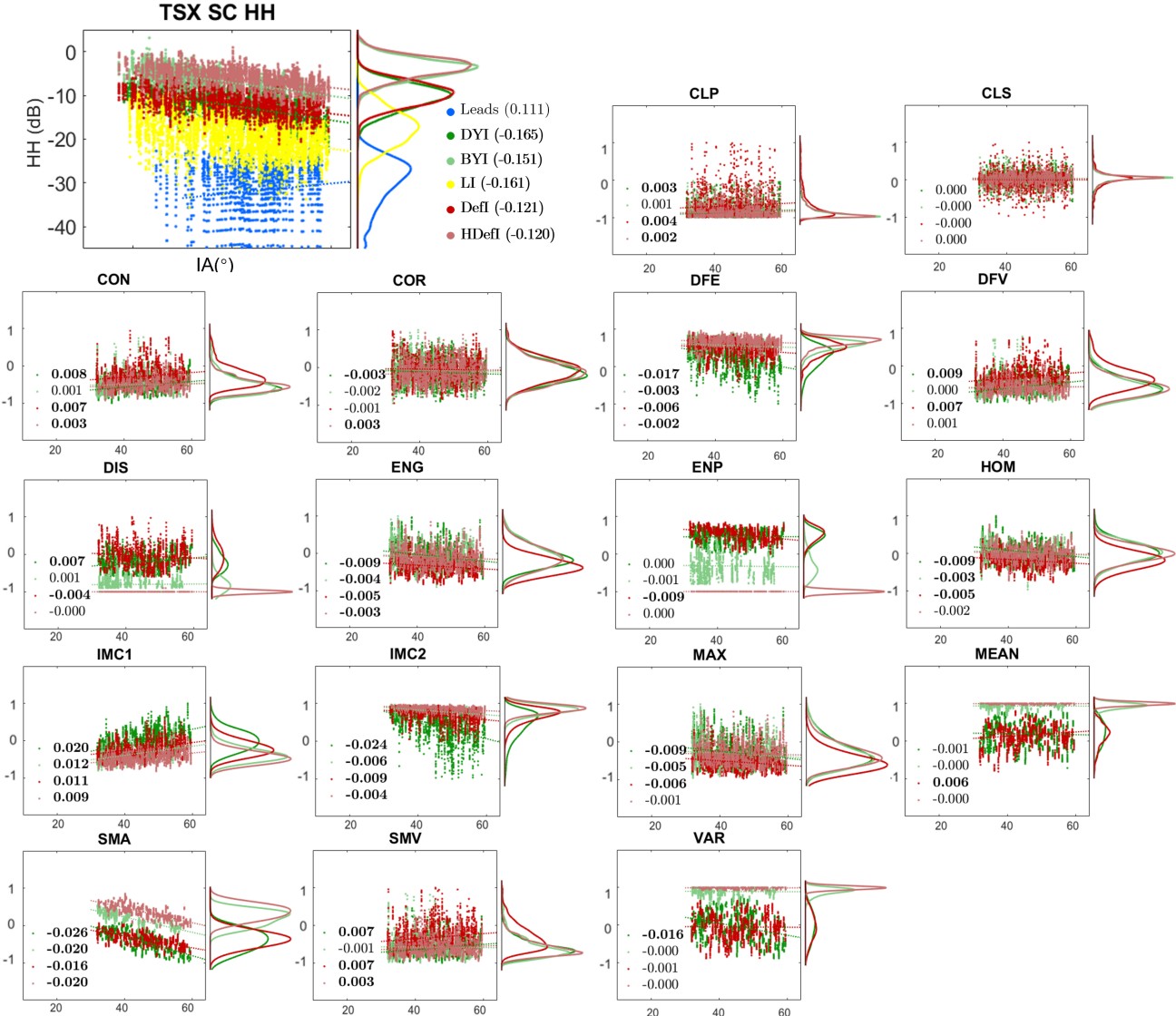

**Figure 4.** Histograms and scatter plots of TSX SC HH intensities with IA for the reference polygons (all classes), and GLCM textures (Table 2) with IA for the training polygons (difficult class pairs). Slope values of different classes with IA are also shown, where the bold font indicates statistical significance. Values of all texture measures are scaled to the -1 to 1 range to yield comparable slope values.

sequent texture-based classification. A separate classification is run using HH intensities only, from which leads are extracted and used for the final classification result, which we found to provide satisfactory lead separation.

IA slopes of C-band and X-band SAR intensities for sea ice types derived in previous studies are shown in Fig. 5. There are a limited number of studies reporting IA dependencies of Arctic sea ice types for X-band sensors. IA slope values shown in Liu et al. (2016), presented in blue asterisks, are derived from TSX SC and WSC scenes with a limited IA range of 22.61°to

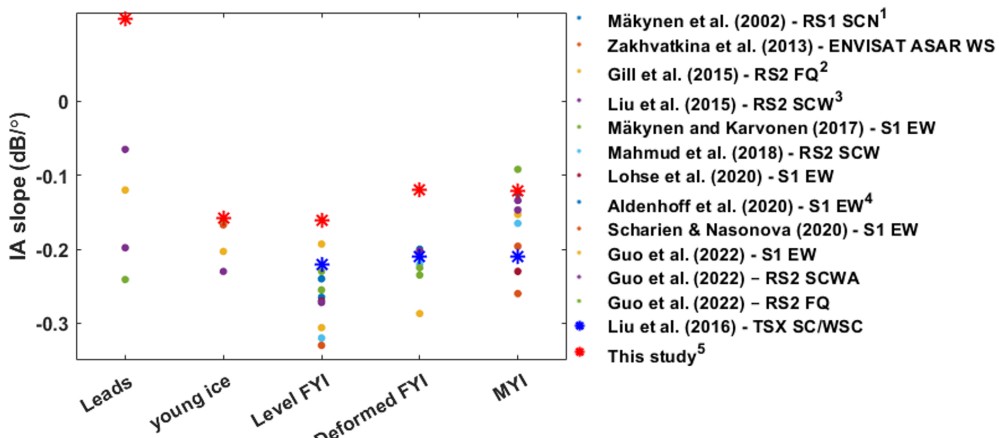

**Figure 5.** Comparison between IA slope values derived in this and previous studies. Different classification schemes are used for these studies, and we summarize them into 5 commonly used classes, and display per-class IA slopes for their classes that are the most closely related to these 5 classes. Dots are for C-band results, and asterisks for X-band ones. Correspondence between ice classes shown in the figure and closest ice classes in the original studies defined differently or more specifically: [1] FYI: FYI with dry snow on top; [2] FYI: land-fast smooth FYI with thin (7.7±3.9 cm) to thick (36.4±12.3 cm) snow cover; [3] Leads: nilas; YI: deformed gray ice; [4] MYI: averaged for MYI and old MYI; [5] YI: averaged for DYI and BYI; Deformed FYI: HDefI; MYI: DefI.

45.31°from the east coast of Antarctica. HH intensities of TSX SC data derived in our study are generally less dependent on IA than those for C-band sensors, the values of which are summarized in Guo et al. (2022)). This is also observed in previous comparative studies of airborne X- and C-band sensors (e.g., Mäkynen and Hallikainen 2004). The general pattern of comparative IA slopes between classes is similar for C- and X-band: LI has a slightly stronger IA dependency than deformed MYI and FYI (in this study HDefI and DefI), presumably due to stronger volume scattering and added randomness in backscatter caused by deformation features, respectively, which both lead to decreased sensitivity to IA (Mäkynen et al., 2002; Dierking and Dall, 2007; Zakhvatkina et al., 2013). These differences confirm the necessity of per-class IA correction in classifying the time series.

## 2.4 Parameter optimization of GLCM textures

Optimization of the above mentioned 3 parameters is performed to provide class separability while maximally retaining spatial details and minimizing the correlation between textures. Texture window size and the combination of texture measures determine the spatial domain for texture calculation and the variety and abundance of GLCM-based statistics used for classification. The co-occurrence distance determines the spatial displacement of gray-level co-occurrences captured by the GLCM and directly impacts resulting texture values.

In this study, class separability is evaluated by the K-S distance (Massey Jr, 1951), which is non-parametric and thus is a relatively robust metric without assumptions of class distribution (Daniel, 1990). The K-S distance quantifies the distance

between class distributions, and the K-S test yields a test decision for the hypothesis that two classes come from the same distribution. Detailed steps of parameter optimization are as follows:

1. for each texture, K-S distances between class pairs are calculated for odd window sizes between 3 and 61 pixels with co-occurrence distances of 1, 2, 4 and 8 that are smaller than the window sizes for all pixels within the training polygons (training pixels);

2. for each combination of textures, the smallest window size at which all individual constituting textures provide statistical separability between all class pairs (as evaluated by the K-S test) for at least one co-occurrence distance is selected as the 'optimal' window size;

3. for each texture combination at its optimal window size and associated co-occurrence distances providing separability between all class pairs, the summation of K-S distances for all textures is divided by the summation of correlation coefficients between texture pairs, resulting in a 'combination rating' that provides control over textural collinearity;

4. texture combinations with the 10 highest ratings in corresponding optimal window size and co-occurrence distance are used to classify the training scenes. The results are compared visually to arrive at a final selection of texture parameters.

GLCM texture calculation for the training pixels and the optimization of texture parameters are conducted in MATLAB 2021b (The Mathworks Inc., 2021). GLCM textures calculated for whole TSX images are then produced with optimized parameters using ESA SNAP and GEE.

## 2.5 Classification of MOSAiC winter time series

Sea ice classification of the time series is conducted using the GIA classifier trained with HH intensities and textures with optimal parameterization. Details of the training process can be found in Lohse et al. (2021). Within the classification process, a Markov Random Field (MRF) contextual smoothing component (Doulgeris, 2015) is added to alter the posterior class probabilities yielded from the classifier before determining maximum probability class labels. This technique replaces global class probabilities with spatially varying local probabilities by giving more weight to class memberships of spatially neighboring classes. This process reduces scattered misclassified pixels caused by texture-based classification and ScanSAR image artifacts, including scalloping and inter-scan banding. These artifacts are are small in areal coverage but wide-spread, thus necessitating a smoothing process. As the area surrounding the CO is the main focus of MOSAiC sea ice studies, we present classification results for a $70\,\mathrm{km}\times70\,\mathrm{km}$ square around the CO.

## 3 Results and discussion

In this section, we first present qualitative and quantitative evaluation of the performance of our classification product. Then, we compare the classification maps with sea ice roughness estimates from MOSAiC in-situ data, and accordingly evaluate our classification scheme splitting FYI and MYI into different deformation states. To evaluate the consistency of the classification, temporal development of areal fractions of each class is then presented and compared with indicators of ice openings from

in-situ data and other MOSAiC studies. Finally, we list several limitations of our workflow and give potential directions for future studies following this work.

## 3.1 Classification with HH intensities and textures

The selected optimal combination of textures used for classification is: DIS, ENG, ENP, HOM, MAX, SMA, and VAR, with an
optimal window size of 9 pixels and a co-occurrence distance of 2 pixels. Fig. 6 shows the comparison between classification results for three example scene subsets using HH intensities only and HH intensities and textures with and without MRF contextual smoothing.

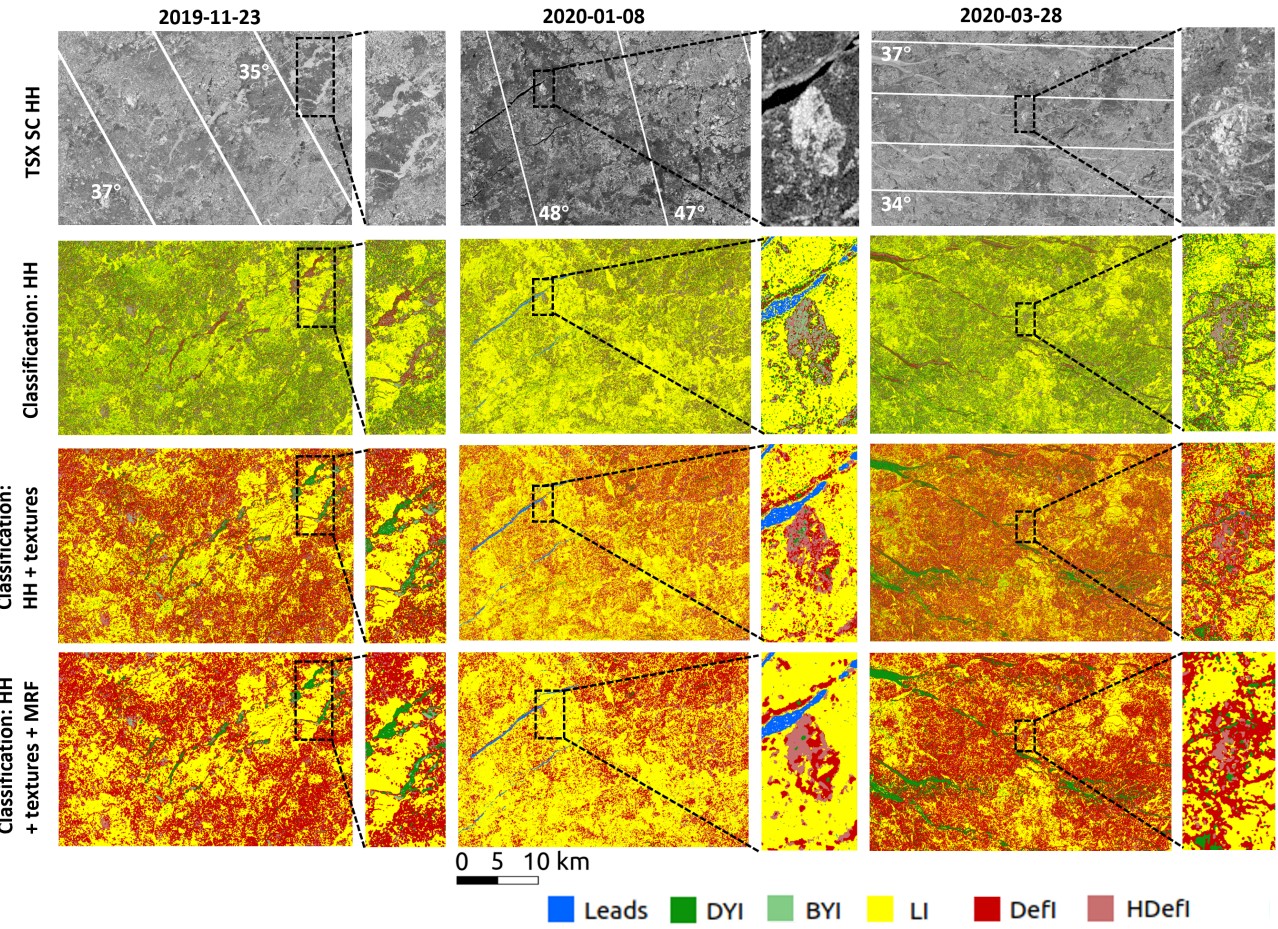

**Figure 6.** Example TSX SC scenes and classification maps using the GIA classifier trained with HH intensities only, HH intensities and the optimal texture measures, and additionally with MRF contextual smoothing applied for 3 scenes across the time series. IA contours are shown as white lines on HH intensities.

Due to ambiguities in HH intensities, classification without textures shows prevalent mixing of difficult class pairs. DefI and HDefI are frequently misclassified as young ice (e.g., 2020.01.08 and 2020.03.28, zoomed-in image patches), resulting in classification maps dominated by DYI and BYI (green). Young ice is also frequently classified as DefI or HDefI (e.g., 2019.11.14, zoomed-in image patch). Considerable classification improvement is achieved from the inclusion of GLCM textures, especially in the correct separation between these class pairs. MRF contextual smoothing further greatly reduces scattered misclassified pixels due to texture classification and image artifacts.

Overall classification accuracies for different testing scenes are shown as box-plots in Fig. 7. The average overall accuracy for the classification of HH intensities and textures (78.31%) is significantly higher (p-value $< 0.01$) than that of HH intensities only (64.79%). The use of MRF contextual smoothing further increases (p-value $< 0.01$) the overall accuracy to 83.70%. For the final classification with MRF contextual smoothing, the confusion matrix (not shown) indicates that remaining misclassifications mostly happen between the difficult class pairs, as expected. Leads and level ice are mostly correctly classified. The MRF contextual smoothing technique is theoretically (Doulgeris, 2015) and practically (not shown) superior to image smoothing processes that do not incorporate contextual information, e.g., a local majority filter, in improving classification accuracy and minimizing the loss of spatial detail.

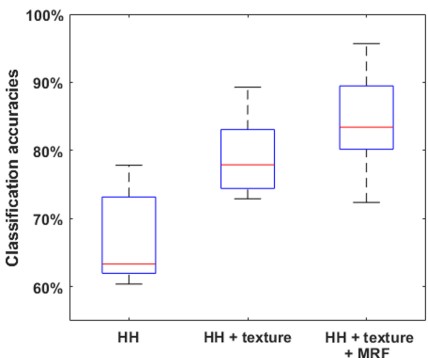

**Figure 7.** Overall accuracies for sea ice classification derived from testing scenes based on TSX SC HH intensities, and HH intensities and GLCM textures with and without MRF smoothing.

To demonstrate temporal classification consistency, classification maps in the middle of each month and the last scene of the time series (2020.03.28) are shown in Fig. 8. The general distribution of LI vs. DefI and HDefI is consistent through the time series for the classified scenes and the MOSAiC ice floe carrying the CO (zoomed-in patches). The classification maps clearly capture the break up and change of size and shape of the MOSAiC ice floe. Major lead openings are seen on 2020.03.17 and 2020.03.28, which are partially classified as BYI and DYI. Panoramic photos taken from Polarstern (Fig. 9(b), Marcel et al. 2021) confirm the presence of ice openings occupied by young ice with the same relative positioning to the ship as indicated by Fig. 8, with Polarstern circled in black in the zoomed-in patches.

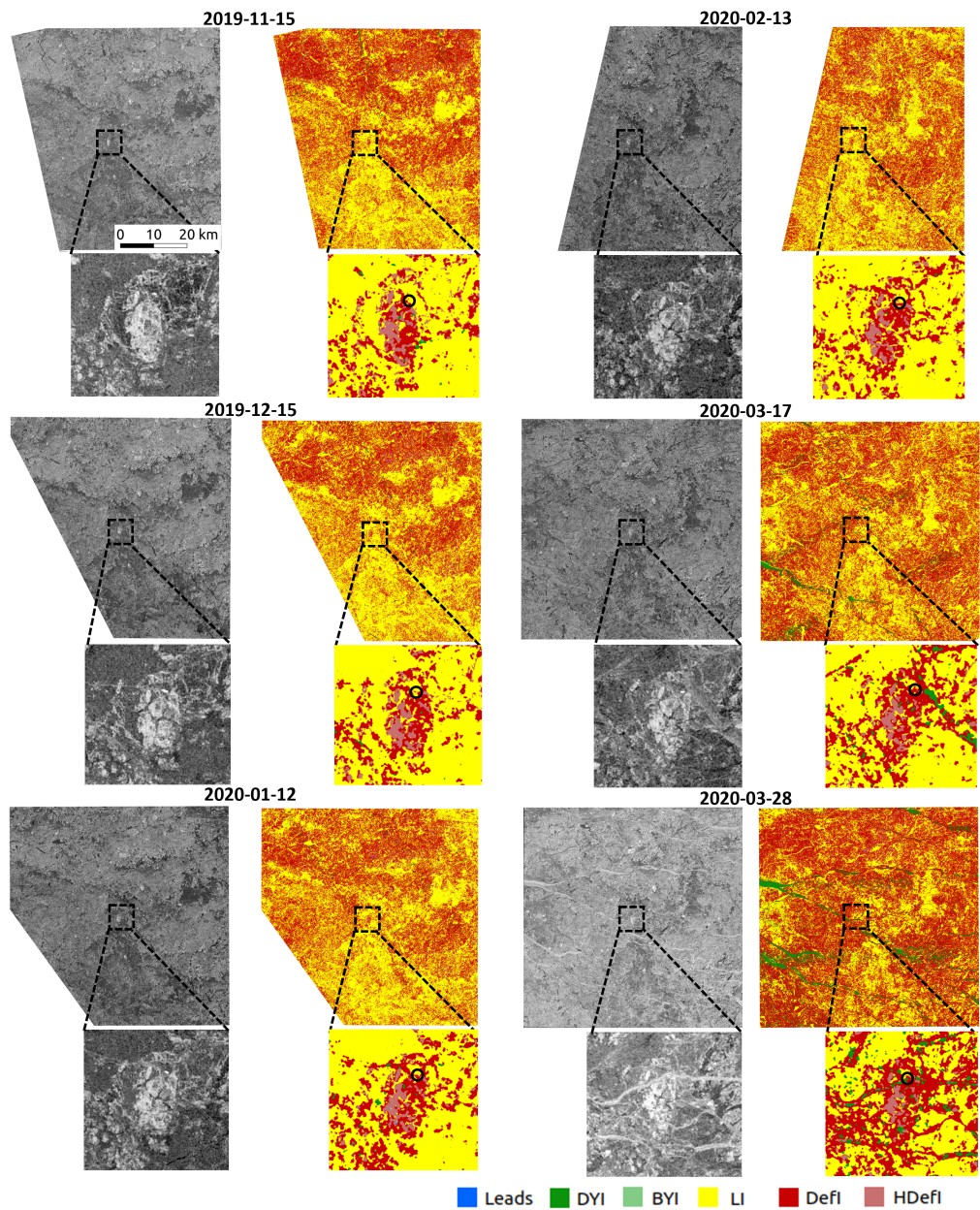

**Figure 8.** Classification maps of example TSX SC scenes, with zoomed-in patches focusing on the MOSAiC ice floe, where black circles indicate the position of Polarstern.

A manual classification map of a small area around Polarstern is shown in Fig. 9(a), which is produced by a co-author with extensive knowledge of sea ice conditions in MOSAiC. Our classification is consistent with ground observations summarized in this map, which indicates that the MOSAiC ice floe was composed of a mixture of FYI and second-year ice (SYI), with

a strongly deformed zone in the center (named 'the Fortress'), which is the oval-shaped ice surface consistently classified as DefI and HDefI (Krumpen et al., 2020; Itkin et al., 2022). In most scenes in November 2019, part of the SYI surface in the MOSAiC ice floe surrounding the Fortress appear similar to or even darker than nearby LI (Fig. 8), thus classified as LI. This

is attributable to the presence of re-frozen melt ponds (Fig. 9(a); Krumpen et al. 2021).

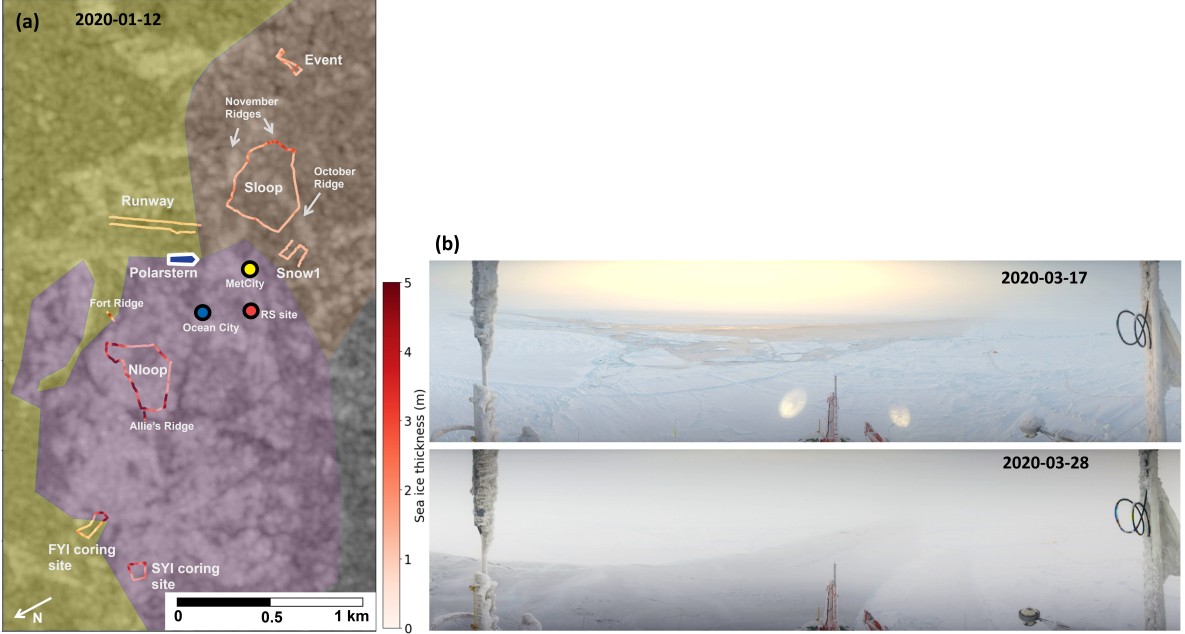

**Figure 9.** (a) Manual sea ice classification of the CO overlaid on a RS2 SCW scene (HH) on 2020.01.12 - yellow: FYI; purple: rough SYI; red: ponded SYI; Polarstern, weather stations, and transects with sea ice thickness measurements are also shown; (b) Panoramic photos taken from Polarstern on 2020.03.17 and 2020.03.28.

## 3.2 Comparison to sea ice roughness estimates

The standard deviation of sea ice thickness measured from the electromagnetic induction (EM) instrument (GEM-2, Hendricks et al. 2022) along several transects near the CO is used as a combined indicator of ice surface and bottom roughness and is plotted in blue on the classification maps in Fig. 10(a). This dataset captures geometric roughness on the sea ice surface

on the spatial scale of ice blocks or pressure ridges, i.e., approximately $1\,m$ to $30\,m$. Geographical coordinates from satellite images and ground data are converted to a local coordinate system that corrects for sea ice drift during data collection (Itkin et al., 2022). Additional minor manual translations are applied to account for geo-location errors. Still, some effects of ice floe rotation and deformation are present, and the data points are averaged in windows of $4 \times 4$ TSX SC pixels (thus $33\,m \times 33\,m$) to partially remedy these issues. Rougher ice (deeper blue) along the transects mostly corresponds correctly to areas classified

as DefI or HDefI and smoother ice (lighter blue) to LI (Fig. 10(a)).

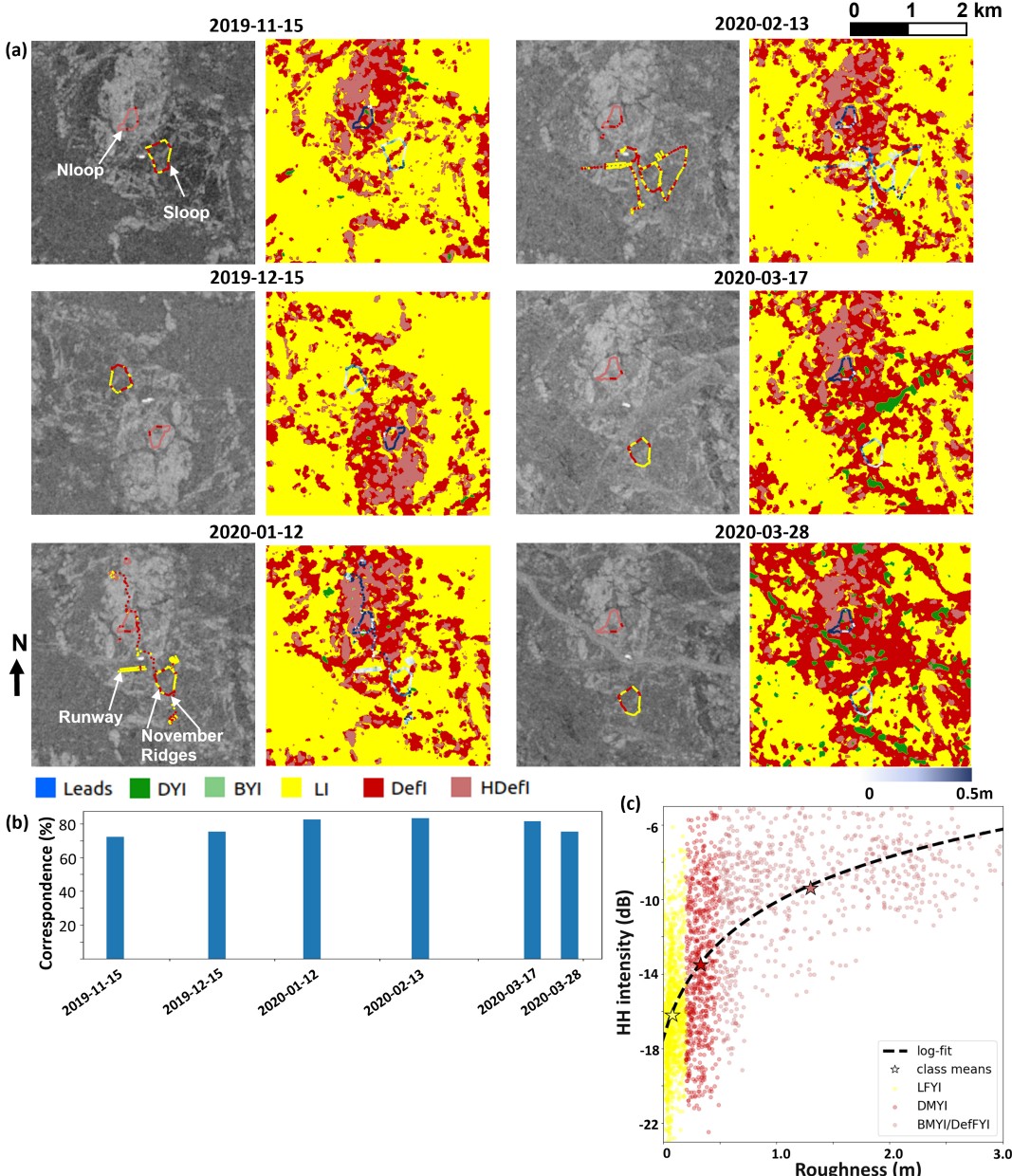

**Figure 10.** (a) Sea ice roughness in transects (in blue) overlaid on SAR classification maps, and classification of sea ice roughness in the same color scheme overlaid on HH intensities, in several transects within a 4 km×4 km square around Polarstern for the same dates as Fig. 8; (b) percentage of correspondence between SAR and roughness classifications in repeated transects; (c) scatter plot of HH intensities vs. ice roughness in repeated transects, grouped by their corresponding class labels from the SAR classification.

Additionally, ice roughness transect points are classified into LI, DefI and HDefI, shown in Fig. 10(a) on top of HH intensities in the same color scheme as the SAR classification. This roughness classification is based on threshold values for level, rubble and ridges ice as described by Itkin et al. (2022). Briefly, in areas of mostly smooth FYI and SYI (outside the Fortress), ice roughness is classified with a threshold of 0.2 m into LI and DefI, with good correspondence with LI and DefI in the SAR classification. In the Fortress, ice roughness is classified using the same threshold into DefI and HDefI, again showing similar spatial distribution to DefI and HDefI in the SAR classification.

Two of the transects are repeated during the entire season: the Southern and Northern transect loops, or 'Sloop' and 'Nloop' (Fig. 10(a)). Sloop is located in the aforementioned ponded SYI area, and crosses rough SYI and smooth refrozen melt ponds, which have similar HH intensities to LI, while Nloop is located in the Fortress and thus consists of predominantly heavily deformed SYI (Itkin et al., 2022). These observations are mostly correctly shown in both the SAR and roughness classifications. The transect 'runway' (Fig. 10(a)), established on LI, is also consistently classified as LI in both classifications.

The transect ice roughness estimate represents surface and bottom roughness, while the SAR classification represents only surface roughness. Consequently, there is an apparent mismatch between the two classifications which can be seen e.g., in Sloop on the 'November ridges,' as pointed out by arrows in Fig. 10(a), most notably on 2020.01.12. In the southern part of Sloop (arrow to the right), where ice is smooth but thick, low HH intensities lead to the SAR classification result of LI, but the roughness classification result is DefI, presumably due to the dominance of ice-bottom roughness. On the contrary, in the western part (arrow to the left), high HH intensities and hence rough ice surface lead to a SAR classification of DefI, while the roughness classification result is mostly LI, likely due to low thickness standard deviations calculated from thin ice (Fig. 9(a)).

The percentages of correct correspondence between both classifications in repeated transects is shown in Fig. 10(b). Corresponding to a roughness classification data point, the SAR classification is counted as 'correct' if the ice class of at least one TSX pixel in its surrounding $4 \times 4$ pixel window reports the same class. Good correspondence is found between the two classifications, with the percentages of correctly classified SAR pixels being consistently near or over 80%.

Finally, we demonstrate the relationship between HH intensities and ice roughness for the repeated transects, grouped by class labels from the SAR classification Fig. 10(c). An apparent logarithmic fit can be seen, where the points representing mean roughness and intensities for LI, DefI and HDefI (shown in stars) are very close to the fitted curve. This indicates that TSX SC HH intensities in these particular transects are controlled largely by geometric ice roughness because other sea ice surface properties, e.g. micro-roughness (centimeter to decimeter), salinity, snow, etc., are quite similar. This relationship, together with the good correspondence between the SAR and roughness classifications shown by Fig. 10(c) and the above qualitative comparisons, illustrates that under similar environmental conditions TSX SC HH intensities of FYI and MYI can be attributed to different degrees of deformation, which justifies our chosen classification scheme. Previous studies have also found similar logarithmic relationship between geometric surface roughness and SAR backscatter intensities for winter sea ice for co- and cross-polarization channels of C- and L-band SAR sensors (Cafarella et al., 2019; Segal et al., 2020), while micro-scale roughness has a more significant impact on C-band backscatter than L-band (e.g., Dierking and Dall 2007; Gegiuc et al. 2018). It is expected that TSX signal, with a shorter wavelength than C-band sensors, should also react to small-scale roughness, but quantifying the contrast between the influence of different spatial scales of surface roughness on SAR backscatter is not

achievable in (with observations used here) nor within the scope of this study. The contribution from both surface and bottom roughness to our ice roughness estimate plus the additional influence from small-scale surface roughness presumably lead to the relatively wide spread of the scatter plot.

### 3.3 Temporal development of ice class fractions

Areal fractions of different classes for all scenes in the time series are shown in Fig. 11. Leads, DYI and BYI are combined into a 'lead ice' category, representing areas of ice opening. DefI and HDefI are combined into a 'deformed ice' category. Relative proportions of level vs deformed ice are reasonably consistent through the time series (Fig. 11). Several peaks of lead ice fraction are visible, most notably in mid-late November 2019 and late January to early February 2020. In March 2020, lead ice fractions remain high when ice openings can be consistently observed in the scenes. A major ice opening event occurred

on 2020.03.28 (Fig. 8) when the lead ice fraction reaches 7.16%. This event persisted through early April.

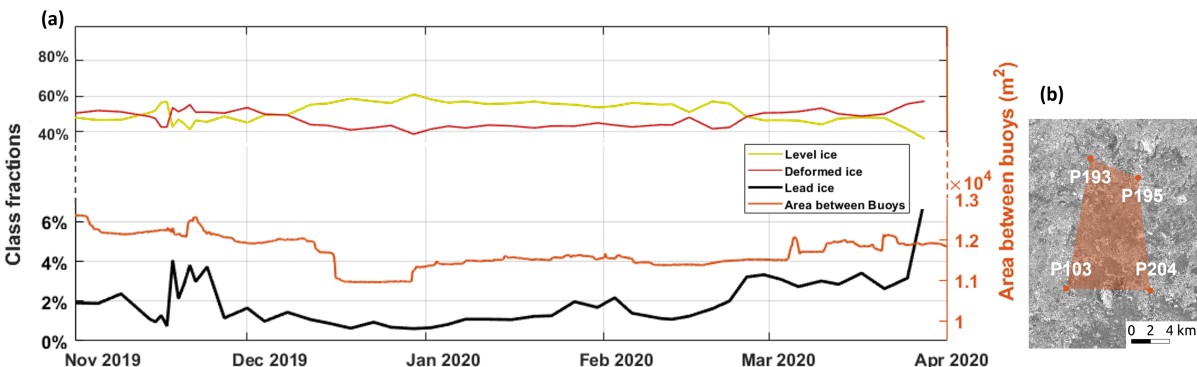

**Figure 11.** (a) Fractions of level ice, deformed ice and lead ice (left y axis), and areas between 4 buoys surrounding the CO (right y axis) over the time series; (b) names and positions of the 4 buoys shown for 2019.11.01.

A detailed examination of ice opening events is conducted by comparing the class fractions to indicators of ice openings derived in this and other MOSAiC studies:

1. Areal change between buoys: the area between 4 selected buoys (P103, P193, P195 and P204, Fig. 11(b)) surrounding the CO (Bliss et al., 2021) are calculated every 3 hours, partially representing events of divergence and convergence (Fig. 11(a),

orange). Similar peaks in this areal change can be seen in mid-late November to those of the lead ice fractions (Fig. 11). Areal changes are also frequent through March, indicating frequent short-lived ice openings between the buoys. The sharp decrease in the area on 2019.12.15 is caused by a large-scale shearing event lasting through 2019.12.23, which is not prominently registered by changes in the lead ice fraction in the scenes due to significantly larger spatial scales.

2. Other MOSAiC studies: several peaks of lead ice fractions in Fig. 11(a) have good correspondence to those generated

from optical satellite observations shown in Krumpen et al. (2021). Lead fractions within a 50 km radius from the CO show prominent peaks in early-mid December, early February and early and late March, matching those in Fig. 11. In the same

study, no lead fraction is produced for mid-late November and mid-late March, but prominent divergence-convergence events can be seen in mid-late November and late March as obtained from S1 sea ice drift data (Krumpen et al., 2021). Similar to our study, a recent sea ice classification study on TSX dual-polarization StripMap images (SM, 54 km×16 km at 3.5m resolution) also identifies a prominent rise in young ice fractions in the 3 km×3 km area around Polarstern in late November 2019 and late March 2020 (Kortum et al., 2022). Abrupt and prominent changes in wind speed and direction were recorded from Polarstern in these periods (Itkin et al., 2022), which likely contributed to the observed increase in lead opening events.

These comparisons demonstrate that our classified time series is valuable as an indicator of ice openings, and thus a good reference to studying associated physical processes in a larger spatial scale than the previously derived MOSAiC sea ice classification product (Kortum et al., 2022).

### 3.4  Limitations and future steps

The current classifier has limited capability in detecting linear young ice areas that are narrower than the texture window size. This is an inevitable outcome of texture-based classification, which we try to mitigate by minimizing texture windows. Comparatively, the leads class is mostly fully represented in the classification, as it is classified with HH intensities only. The texture parameter selection workflow established in this study produces satisfactory classification results (Section 3.1 and Section 3.3) and is generally applicable for future studies. However, the texture parameters yielded are specific to our dataset on the constrained IA range of the training scenes.

The inherent scalloping and inter-scan banding issues in ScanSAR products can be observed in HH intensities and textures with varying visibility across scenes and are more prominent in HH textures than in HH intensities. These image artifacts affect the classification results, most notably leading to misclassification between the difficult class pairs. This issue is partially remedied by MRF contextual smoothing. For future studies using TSX SC scenes with obvious sensor artifacts, additional correction steps should be taken using previously proposed procedures (e.g., Iqbal et al. 2012; Yang et al. 2020).

No continuous *in situ* observation is available to provide detailed information on thin ice evolution through the time series. Ice roughness derived from *in situ* ice thickness measurements is calculated on a different spatial scale than the classification maps, represents both surface and bottom ice roughness, and suffers from potential co-location errors due to sea ice rotation and deformation. Therefore, the utilization of ice surface roughness calculated from air-borne and ground-based laser scanners is desirable in future studies as a stronger validation of ice classification. This study has focused on the freezing season during the MOSAiC expedition. Future steps will extend the study period into summer to examine the seasonality of TSX SC textures on sea ice and its effects on texture-based sea ice classification.

For future studies on texture-based sea ice classification, more detailed quantification of the correspondence between GLCM textures and ice surface properties should be conducted, following previous studies (e.g., Baraldi and Parmiggiani 1995; Soh and Tsatsoulis 1999). Also, previous studies of texture-based sea ice classification have achieved ice type separation in various physical window sizes. Therefore, investigation into better including multi-scale textural information, e.g., by varying window size and co-occurrence distance, is desirable (e.g., Soh and Tsatsoulis 1999; Leigh et al. 2014). Although GLCM textures are among the most powerful tools for texture-based classification (Hall-Beyer, 2017; Zakhvatkina et al., 2019), it is still valuable to

examine IA dependencies and the utilization of other types of image textures previously used for sea ice classification, e.g., first-order textures, image moments, and MRF-based, wavelet transformed-based, variogram-based, and Gray Level Dependence Matrix (GLDM) based textures, etc. (Conners and Harlow, 1980; Unser, 1995; Clausi, 2001; Clausi and Yu, 2004; Sanden and Hoekman, 2005; Bogdanov et al., 2007; Komarov and Buehner, 2017; Gegiuc et al., 2018; Scharien and Nasonova, 2020). Finally, the integration of ice-type specific IA dependencies into other classifiers, e.g., Convolutional Neural Network (CNN) based classifiers (e.g., Boulze et al. 2020), is desirable to potentially improve classification performance.

## 4   Conclusions

This study demonstrates per-class IA slopes of HH intensities and GLCM textures calculated from TSX SC data, and uses a sea ice classifier incorporating these IA dependencies to produce a classified time series for winter MOSAiC. Linear IA dependencies of HH intensities in dB in our study area and period are generally lower than C-band data, but between-class IA slope differences still necessitate per-class IA correction. In the constrained IA range, GLCM textures calculated from dB intensities also exhibit linear IA dependencies. The leads class has a wide scatter in HH intensities and textures vs. IAs resulting in weak linear dependency, and is thus retrieved from a separate classification on HH intensities only. A texture parameter selection process based on statistical separability between class distributions determines the optimal texture combination to be DIS, ENG, ENP, HOM, MAX, SMA, and VAR (see Table 2 for definitions) at a window size of 9 pixels with a co-occurrence distance of 2 pixels. We use a classification scheme that separates young ice into different SAR intensities and FYI and MYI into different deformation states. Qualitative assessments through visual inspection of classification maps and quantitative assessment using classification accuracies show that the inclusion of GLCM textures is essential in classifying single-polarization TSX SC data. The application of MRF contextual smoothing refines the result while preserving maximum spatial details, leading to significantly increased classification accuracies. Good correspondence is found between the classification result and geometric ice roughness calculated from *in situ* ice thickness measurements, the latter showing a logarithmic relationship with HH backscatter intensities. The classified time series show reasonably consistent fractions of LI vs. DefI and HDefI through the time series. Lead ice fractions derived from the classification result correspond well with other indicators of ice openings derived in this and previous studies. This suggests that the classified time series can serve as a reliable reference of the sea ice conditions and associated physical processes during the expedition within the spatial scale of TSX SC scenes. This study provides valuable information on the utilization of per-class IA dependencies of TSX SC intensities and GLCM textures in classifying sea ice and a classification product of a broad area surrounding the MOSAiC ice camp that can potentially facilitate future MOSAiC sea ice studies and modeling efforts.

*Data availability.* The classified time series presented in this publication is available as projected GeoTIFFs (in EPSG:3575) here: https://www.dropbox.com/sh/edx4eq2oux0fqdg/AAB5CXZ8ReTwZNpXe48mpoZYa?dl=0. An updated version of the classified time series with

a wider temporal and spatial coverage will be published in the PANGAEA platform https://www.pangaea.de/. Correspondence between pixel values and class labels: 3: leads; 5: DYI; 6: BYI; 7: LI; 9: DefI; 10: HDefI.

*Author contributions.* PI, SSI, APD, MJ, and GS were involved in project administration and supervision. All co-authors were involved in the conceptualization of the study. SSI was responsible for TSX SC data acquisition. WG was responsible for data curation, methodology designing, formal analysis, and result visualization. PI conducted the analysis on sea ice surface roughness using sea ice thickness measurements from MOSAiC, and provided the manual sea ice classification map surrounding the CO and the time series of areal change between the buoys. APD provided his codes and knowledge of MRF contextual smoothing. WG prepared the manuscript, with contributions from all co-authors in reviewing and editing.

*Competing interests.* No competing interests are present.

*Acknowledgements.* The authors would like to thank all persons involved in the expedition of the Research Vessel Polarstern during Multi-disciplinary Drifting Observatory for the Study of the Arctic Climate (MOSAiC) in 2019–2020 as listed in Nixdorf et al. (2021). This work was supported by the Research Council of Norway (RCN) projects: Sea Ice Deformation and Snow for an Arctic in Transition (SIDRiFT) (287871), Center for Integrated Remote Sensing and Forecasting for Arctic Operations (CIRFA) (237906), and Project Oil spill and newly formed sea ice detection, characterization, and mapping in the Barents Sea using remote sensing by SAR (OIBSAR) (280616). SSI and GS are supported by the Deutsche Forschungsgemeinschaft (DFG) through the Project "MOSAiCmicrowaveRS" (Grant 420499875).

Data used in this article were produced as part of MOSAiC with the tag MOSAiC20192020 and Project_ID: AWI_PS122_00. TerraSAR-X images used in this study were acquired using the TerraSAR-X AO OCE3562_4 (PI: SS). RADARSAT-2 data was provided by NSC/KSAT under the Norwegian-Canadian Radarsat agreement 2019 and 2020. Sentinel-1 data is publicly available from the Copernicus Open Access Hub (https://scihub.copernicus.eu/, last access: Oct 2021; European Space Agency 2021). MetCity temperature data were provided by National Science Foundation project OPP-1724551 (Cox et al., 2021). The OSI SAF global sea ice type product (OSI-403-d) is publicly available from https://osi-saf.eumetsat.int/products/osi-403-d (last access: Oct 2021; OSI SAF 2019). The NSIDC IST dataset (MOD29/MYD29) is publicly available from https://nsidc.org/data/MOD29 (last access: Oct 2021; Hall and Riggs. 2021).

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
