# Peer review of "Sea ice classification of TerraSAR-X ScanSAR images for the MOSAiC expedition incorporating per-class incidence angle dependency of image texture"

_The Cryosphere, 2022_

## Referee Comment (RC1)

The manuscript "Sea ice classification of TerraSAR-X ScanSAR images for theMOSAiC expedition incorporating per-class incidence angle dependency of image texture" presents methodology and results of sea ice type classification of TerraSAR-X imagery obtained during the MOSAiC expedition. Despite very interesting findings, due to large diversity of methods, results and analysis and a large size of the manuscript, it is recommended to split the manuscript in two parts, improve the order of the presentation and resubmit the manuscript(s) after a major revision.

**Major comments**

Although only two objectives are formulated in the introduction, the impression is that the manuscript attempts to fulfil at least four: 1. Investigate per-class AI dependence; 2. Optimize parameters of texture features; 3. Train and evaluate classifier; 4. Analyse time series. In my opinion such variety of objectives does not allow to focus well. That makes the manuscript too long to read and the story too difficult to follow. I would suggest to completely remove section 3.3 and correspondingly reduce section 3.4. I'm confident that results shown in these sections deserve a separate paper. I will therefore focus my review on the first, methodological part.

What is GIA classifier? Authors refer to that term in many places, but it is never defined or explained. I guess, that's one of the central blocks in the classification algorithm: apparently, the backscatter, the texture features, the IA are passed into the mysterious "GIA classifier" for doing the actual classification. But how?! I'm very curious to know. GIA classifier needs to be clearly explained.

Order of presentation needs to be revised in order to correspond to the selected logic (Intro, Data and methods, Results and Discussion): Lines 133 – 144 and Fig. 4 should come in Section 2.1 Data; Lines 151 – 164 with Table 1 and lines 271 – 275 with Table 2 belong to Introduction as they describe state-of-the-art; Section 2.3.2 belongs to Results as it describes WHAT is achieved and not HOW it is achieved.

Analysis of IA dependence for various ice types need to be increased as it is an important result of this work. What is error-bars of the slopes (it can be computed, e.g. by bootstrapping) and what is significance? What is the reason for large positive bias of the slopes – speculation on stronger volume scattering needs to be expanded. What is physical reasoning behind positive slope for leads?

The suggested method and parametrisations seem to be difficult to use in other conditions (C-band, other IA, other ice types, summer). Although it is mentioned as a limitation in the end, I believe it is important to also underline in the Introduction – the goal is to study a specific TSX SC timeseries and for analysis of another dataset a similar full-scale analysis needs to be performed.

Image size and number of texture features are undoubtfully important hyperparameters of the Haralick algorithm. However, neglecting quantisation level and distance to neighbour pixels can lead to significantly worse results. Sensitivity to these two parameters should also be studied, for example in this respect: how does despeckling boxcar filter impact the GLCM? In theory, if a 3x3

filter is applied and then GLCM is computed with 2 pix distance, there should be almost no elements in GLCM off the main diagonal. On another note, Haralick (1973) suggested using adjacent pixels (d=1) so the choice of authors d=2 should be tested and explained better.

**Minor comments**

L7. Phrases in parenthesis make the sentence very unclear. Please split into two sentences.

L12. Unfortunately the GIA classifier and class probabilities are never explained in the manuscript.

L24. Please provide reference to prove the "largest expedition in history"?

L71. Objective 1 is actually two objectives: 1 . to investigate and demonstrate per-class IA dependencies of TSX SC HH intensity and GLCM textures; 2. to determine the feasibility and optimal parameterization of including texture measures as input features to the GIA classifier.

Figure 1 shall be removed as it does not explain anything.

L96. "and shown in details in " -> ", dates shown in"

L107. Why were these ice categories chosen? It should be written that other categories were not present in the dataset and the method cannot be extrapolated.

L129. Polygons == rectangles? This is unclear.

L133. Maybe "evolution of young ice" ?

L135. Please rewrite "wide-spread lead openings of open water or nilas" as "wide opening of leads with water or nilas"

Figure 3. The smallest sub-images seem to be very blurred. Is it the effect of the despeckling filter or just visualisation?

L208. Cannot agree here. Other authors also studied distance and number of grey levels (e.g. Clausi 2002). Sensitivity to these two parameters need also to be studied (see major comments).

L221. "...and thus is a relatively..."

L236 and 237. Is that already results of parameter optimization? Then it is better to move to the Results section.

L268. Whay volume scattering is presumed to be stronger?

Figure 6. Is positive slope for leads even physical? How the strong positive bias of the slopes can be explained?

L274. "This is given that" can be removed.

L276 – 280. This seems to be logical after the results, in the Conclusions section.

L321. A reference to unpublished work just supports my concern that it is too early to include this section in the manuscript.

Figure 10. It is impossible to see shades of blue on the roughness transects.

Figure 11 and Lines 389 - 392. Why the 10% sudden drop of the polygon area on ~15 December is not reflected in a similar change of MY or young ice? Why does the lead ice increases ~3 times on 1 March and this is not reflected in the polygon area? Where are the plots of "other mosaic studies" that are easy to compare with fractions of different ice types? I'm afraid it is too early to write "that the classified time series is valuable as indicator of ice openings" as I cannot see a proof of that. Instead, the variations of ice fractions seem to be rather spontaneous and connected to uncertainty of the algorithm.

L396. "The leads class are mostly fully represented in the classification map" is it really a limitation? Can be removed from that section.

L425. Convolutional neural networks also deserve being mentioned as a potential tool.

---

## Referee Comment (RC2)

Review to submitted manuscript; "*Sea ice classification of TerraSAR-X ScanSAR images for the MOSAiC expedition incorporating per-class incidence angle dependency of image texture*". The manuscript investigates per-class sea ice incidence angle dependencies in TerraSAR-X ScanSAR images and GLCM textures and trains a Bayesian classifier to classify sea ice surrounding the MOSAiC expedition.

Thank you for a well written manuscript with strong English and interesting results covering a high-profile scientific campaign. To summarize the main critique points: the paper is too long, convoluted to read at times, and it is difficult to keep track of discussed subjects. In addition, parts of the methodology needs to be further clarified.

I agree with the other reviewer that the manuscript would be better suited split into two, and resubmitting them with major revisions. One manuscript could focus on the IA dependency of the TSX SC intensity and the GLCM textures, while the other could examine the GIA and the time-series of the MOSAiC campaign.

**Major Comments**

Disclaimer. I have limited practical experience with bayesian classifiers but extensive knowledge of deep learning with emphasis on sea ice segmentation using convolutional neural networks. Reviewing the methodology regarding the bayesian classifier raises the following concerns for me, which I would like you to consider and address:

- Limited testing (validation in your words) examples rectangles
  - 10 reference 3x3 pixels for each class is selected for each reference scene (13 scenes in total). This is a total of 1,170 pixels for each class. Considering the abundance of data at your disposal (>1,000 x >1,000 pixels in each image?), I would refrain from needle picking select small areas. Labelling data is a time consuming task but there are tools available, which could assist, e.g. https://github.com/ESA-PhiLab/iris. At least I would require a justification for the approach.
- Small size of testing rectangles
  - Why are 3 x 3 pixel rectangles selected? Could they be larger? Why not? Do the pixels have to be separate or could you label an area with multiple classes?
- Spatial and temporal biased training and testing
  - Generally training and testing should be carried out on areas without spatial or temporal correlation, i.e. on different scenes to avoid biases spilling over from the training to the testing phase. As the data is randomly split in training and test, I fear that some pixels may lie very close together, and could artificially improve the model performance but without carryover to generalization of the classifier (i.e. may not be as reliable on non-testing data).

More information on how the classifier is trained should be included. How is it optimized?

**Data**

The data selection should be more clearly explained or alternatively visualized using the acquisition dates. 53 scenes are used in this study, 50 during the MOSAiC campaign, 3

afterwards with low IAs to complete the spectrum. 40 of these scenes are not used for training the classifier (as I understand it). 13 of the 53 scenes have 10 3x3 rectangles labelled and used for training and testing.

Generally, when optimizing models, data is typically split into training, validation and testing and if supervised methodologies are applied, each split will have raw data (X) and a reference (Y), i.e. the "ground truth". Typically, a validation subset should be utilized for decision making during the optimization process, i.e. should we stop (early stopping), should we tweak the learning or regularization parameters? And finally the model performance is evaluated on the test data, which no optimization changes have been made upon. As I understand the GIA training process, you are using a test subset, and should call it as such.

In regards to the segmentation tools applied, personally, I would have chosen to apply convolutional neural networks. At least it should be mentioned as a potential area of future work.

**Minor Comments**
L54: Why does the TSX SC data only come in the HH polarization?

L128: 10 reference rectangles of 3 x 3 pixels sounds very small. That is only 90 pixels per class per scene, i.e. 1170 pixels.
Is every class represented in every scene? And how certain are you of your qualitative selection?

L129: Improving consistency between training scenes using a 40 km x 40 km area is unclear to me. How does this work?

L180: Only textures of HH intensities have a consistent relationship with IA.. HH intensities as opposed to what? Or is it referring to the scaling of the image, i.e. dB.

L337: 'On the contrary..' This sentence is quite difficult to read. I think it should be split up into two sentences.

In addition, there is a pdf document attached with grammatical suggests.

[Figure]

[Figure]

**Sea ice classification of TerraSAR-X ScanSAR images for the MOSAiC expedition incorporating per-class incidence angle dependency of image texture**

Wenkai Guo[1], Polona Itkin[1], Suman Singha[2], Anthony Paul Doulgeris[1], Malin Johansson[1], and Gunnar Spreen[3]

[1]Department of Physics and Technology, UiT The Arctic University of Norway
[2]Maritime Safety and Security Laboratory, Remote Sensing Technology Institute (IMF), German Aerospace Center (DLR), 28359 Bremen, Germany
[3]Institute of Environmental Physics, University of Bremen

**Correspondence:** Wenkai Guo (wenkai.guo@uit.no)

**Abstract.**

In this study, we provide sea ice classification maps of a sub-weekly time series of X-band TerraSAR-X ScanSAR (TSX SC, HH polarization) images from November 2019 to March 2020 covering the Multidisciplinary drifting Observatory for the Study of Arctic Climate (MOSAiC) expedition. This classified time series benefits from the wide spatial coverage and

5  relatively high spatial resolution of the TSX SC dataset, classifying sea ice into leads, young ice with different intensities, and thick ice with different degrees of deformation. We use a classification method considering per-class incidence angle (IA) dependencies (the Gaussian IA classifier) to correct the IA effect (decreasing backscatter with increasing IAs) specific to each class. In addition to HH intensities, we use Gray-Level Co-occurrence Matrix (GLCM) textures as input features to aid the task of one-band classification. Accordingly, we investigate and demonstrate IA dependencies of TSX SC intensities and

10  image textures for different sea ice classes, which are found to be generally lower than those for C-band SAR data. Optimal parameters for GLCM texture calculation are derived to achieve good separation between class distributions while keeping maximum spatial detail and minimizing texture collinearity. Class probabilities yielded from the GIA classifier are further adjusted by a Markov Random Field (MRF) contextual smoothing process to generate final classification results. A significant increase in classification performance is achieved from the inclusion of textures with optimized parameters, as evaluated by

15  classification accuracies (final overall accuracy: 86.05%) and comparison to sea ice roughness derived from sea ice thickness measurements (correspondence consistently close to or higher than 80%). Areal fractions of classes representing ice openings (leads and young ice) correspond well with ice opening time series derived from *in situ*, satellite SAR and optical data in this and previous studies. This study provides a SAR perspective on the changing sea ice conditions surrounding the MOSAiC ice camp through the expedition, and a useful basic dataset for future MOSAiC studies on physical sea ice processes and ocean

20  and climate modeling.

[Figure]

**1 Introduction**

During the one-year-long Multidisciplinary drifting Observatory for the Study of Arctic Climate (MOSAiC) expedition from 2019 to 2020, the icebreaker RV Polarstern drifted with sea ice along the Transpolar Drift in the Central Arctic, conducting the largest multidisciplinary Arctic research expedition in history. Satellite data acquisitions from multiple platforms were coordinated to survey the broader sea ice area surrounding the expedition, enabling continuous large-scale monitoring of physical sea ice conditions along the drift. Also, extensive on-ice, airborne and ship-based *in situ* data was collected surrounding the MOSAiC ice floe, where Polarstern was moored and the Central Observatory (CO) established. These include data from meteorological stations, airborne laser surveys, ship radar measurements, and a distributed network of autonomous buoys, etc (Krumpen and Sokolov, 2020; Nicolaus et al., 2021; Shupe et al., 2022). This expedition aimed to facilitate detailed physical, biogeochemical and ecological studies of the region, enabling multi-scale quantification of relevant processes and feedbacks and eventually the production of improved climate and Earth system models (Krumpen et al., 2021; Nicolaus et al., 2021; Shupe et al., 2022). The classification of sea ice types is an important basic representation of sea ice conditions which supports various types of further analyses, e.g., monitoring ice break-up and lead formation, inferring the occurrence of sea ice deformation, studying ice-associated and under-ice ecology, and as input to ocean and climate models, etc.

Satellite Synthetic Aperture Radar (SAR) data has been used widely for sea ice classification for operational and scientific purposes due to its high spatial resolution and weather- and illumination-independent monitoring capabilities (Zakhvatkina et al., 2019). Coordinated acquisitions of TerraSAR-X ScanSAR (TSX SC) data were conducted to specifically provide consistent coverage of the MOSAiC ice floe throughout the expedition. This dataset provides daily X-band (9.65 GHz) imaging with 8.25 m nominal pixel spacing (considerably higher than open-access ScanSAR products, e.g., Sentinel-1 (S1)), and an extent of approximately 100×150 km, and is a valuable data source for long-term examination of sea ice development for MOSAiC. This study aims to produce a classified winter (November 2019 to March 2020) time series of TSX SC data surrounding the CO, which can serve as a basis for further MOSAiC sea ice studies and modeling efforts.

TSX SC scenes in this time series cover a wide range of incidence angles (IAs). Therefore, appropriate adjustment for the IA effect of SAR signal (generally decreasing backscatter intensities with IA) is crucial for reliable and consistent classification of the time series. It has been demonstrated that the magnitude of the IA effect varies with sea ice types (Mäkynen et al., 2002; Mäkynen and Juha, 2017; Mahmud et al., 2018). This phenomenon necessitates per-class correction of the IA effect. Therefore, a SAR sea ice classifier which considers between-class IA dependency differences is used in this study. Named the Gaussian Incidence Angle (GIA) classifier, it directly incorporates per-class IA dependencies into a Bayesian classifier, treating the IA dependence as a class property instead of an image property. This is achieved by replacing the constant mean vector of the Gaussian probability density function with a linearly variable mean (Lohse et al., 2020). This classifier has been developed for use with S1 ExtraWide (EW) data, and has also be used with Radarsat-2 ScanSAR Wide and Fine Quad-pol (RS2 SCW and FQ) data with minor adjustments (Guo et al., 2022). The GIA classifier reliably corrects the IA effect on HH and HV channels of these datasets, resulting in improved classification results compared to classification on scenes with global IA correction.

[Figure]

TSX SC data used in this study comes in the HH polarization. The same ice types can have vastly different HH intensities due to different surface characteristics, e.g., different degrees of deformation on FYI and MYI, and different surface roughness and salinity levels on young ice, etc. On the other hand, some ice types have been shown to have similar X-band HH intensities, e.g., deformed first-year ice (FYI), multi-year ice (MYI) and young ice (e.g., Liu et al. 2016). Therefore, in addition to HH intensities, we use Gray-Level Co-occurrence Matrix (GLCM) texture measures as input to the classification to expanded the feature space. In the logarithmic (dB) domain, S1 EW textures of the HH channel for open water and different ice types generally have a linear relationship with IA (Lohse et al., 2021). To our knowledge, no previous study has demonstrated IA dependencies of different Arctic sea ice types for TSX SC intensities and GLCM textures. This study examines the statistical distributions of HH intensities and textures of TSX SC scenes in the MOSAiC winter time series, and evaluates their IA dependencies and hence applicability as input features to the GIA classifier.

The optimal texture window size and set of texture measures to use for texture calculation are derived to provide statistical separability between class distributions, which is evaluated by the Kolmogorov-Smirnov (K-S) distance (Massey Jr, 1951). For better general applicability of our proposed classification workflow, 17 GLCM texture measures are analyzed which are derivable using commonly available software and online tools, i.e., ESA SNAP (European Space Agency, 2020) and the Google Earth Engine (GEE, Gorelick et al. 2017). As we aim to fully utilize the spatial resolution provided by TSX SC data, a rating system is developed to find the set of texture measures that provides separability between classes at the smallest possible window sizes, while minimizing inter-correlations.

In summary, the objectives of this study are: 1. to investigate and demonstrate per-class IA dependencies of TSX SC HH intensity and GLCM textures, based on which to determine the feasibility and optimal parameterization of including texture measures as input features to the GIA classifier; 2. to train the GIA classifier to produce a classified winter time series for the sea ice area surrounding the MOSAiC expedition.

**2 Materials and methods**

Materials and methods used in this study are summarized in Fig. 1, and explained as follows.

**2.1 Data**

This study uses 50 scenes during the MOSAiC winter (2019.11.01 to 2020.03.28, IA: 31.90 to 59.56 ) for the examination of IA dependencies of HH intensities and textures, and sea ice classification (hereafter referred to as the time series), with an average of 3 scenes per week. Additionally, 3 scenes are picked from 2020.03.31 onward (2020.03.31, 2020.04.03 and 2020.04.11, IA: 17.18 to 36.70 ) and are only used for the demonstration of IA dependency of HH intensities, which completes the coverage of the full IA range of TSX SC data. All scenes are radiometrically corrected and calibrated to $^0$ and subjected to a speckle filter (boxcar, 3×3), and then converted to dB. Fig. 2(b) shows the scene boundaries, and Fig. 2(c) shows their IA ranges.

[Figure]

[Figure]

**Figure 1.** Materials and methods.

Scenes after 2020.03.31 were captured at low IAs (Fig. 2(c)) to keep the CO, which was drifting below 85.5 N (Fig. 2(b)), within the scene frames, and exhibit consistent linear IA dependency with other scenes for HH intensities but not for HH textures (not shown). The spatial details obtainable from these scenes are different from others after being subjected to identical pre-processing steps, resulting in considerably different texture values. Additionally, these scenes generally have higher noise floors than the rest of the scenes, and are thus more affected by noise (Fritz et al., 2013). Therefore, they are useful for the investigation of IA dependencies of TSX SC intensities for lower IAs, but not for deriving a consistent texture-based classifier for winter MOSAiC.

Environmental conditions are inferred from 2m air temperature records extracted from the weather station MetCity in the CO (Fig. 2(c)), which show that the temperatures remained mostly below $-5$ C throughout the study period except for late April, when warm spells brought temperatures to near 0 C. A subset of TSX SC scenes are selected in freezing conditions (reference scenes), from which reference polygons are derived for training, validation, and the examination of IA dependencies. A total of 13 reference scenes (plotted in Fig. 2(c) and shown in detail in Fig. 3(b)) are chosen to cover each month between November 2019 and April 2020 and the whole IA range of TSX SC data (17.18 to 59.56 ). Among these, scenes before 2020.03.31 are used for training and validation (training scenes).

[Figure]

For the reference scenes, cloud-free pixels from the National Snow and Ice Data Center (NSIDC) MOD29/MYD29 sea ice
surface temperatures (IST) dataset (Hall and Riggs., 2021) are extracted to show temperatures within the scene boundaries. For
each reference scene, the IST scene with maximum cloud-free overlap ($> 70\%$ of the scene area) within 3 hours of TSX SC
acquisition is manually selected and used to ensure that ISTs are well below -5 (Fig. 2(d)). Overlapping S1 EW and RS2 FQ
scenes and the Ocean and Sea Ice Satellite Application Facility (OSI SAF) sea ice type (OSI-403-d, Fig. 2(a)) product (OSI
SAF, 2019) are used qualitatively as visual reference to aid the derivation of reference polygons, providing general knowledge
about large-scale ice conditions and comparison with C-band SAR signals, respectively.

**2.2 Reference polygons of sea ice classes**

We classify sea ice into leads, rough young ice with different HH intensities, and thick ice (FYI or MYI) with different
roughness levels. This is explained in details below, where intensity thresholds are visually derived approximate values only
used as one of the criteria in deriving the reference polygons:

1. Leads: ice openings occupied by calm open water, nilas or smooth newly formed ice, having the lowest HH intensities
($< -25$ dB). The separation between open water in different wind states is not within the scope of this study, and visual
examination shows that open water leads in the TSX SC time series are all narrow ($\leq 250 m$) and predominantly in a calm
state.

2. Dark young ice (DYI): rough, newly formed ice surfaces in open leads with relatively high HH intensities ($\geq -15$ dB)
are all regarded as young ice, irrespective of ice thickness. Young ice is further split into two separate classes to aid the
classification of single-band TSX SC data. This is only done to account for areas with distinctive difference in HH intensities,
presumably due to the evolving surface roughness, e.g., influenced by the growing and disappearing of frost flowers which
are highly saline and causes changing scales of surface roughness through time, thus strongly impacting X-band SAR signals
(Martin et al., 1995; Barber et al., 2014; Isleifson et al., 2018; Johansson et al., 2018). The separated young ice classes do not
correspond to existing ice types given in the WMO nomenclature (WMO, 2018). The DYI class includes young ice areas with
comparatively low HH intensities (between $-15$ dB and $-10$ dB).

3. Bright young ice (BYI): rough young ice with HH intensities of greater than $-10$ dB.

4. Level ice (LI): smooth thick ice (FYI or MYI) areas having intermediate HH intensities, between leads and DYI ($-25$ dB
and $-15$ dB).

5. Deformed ice (DefI): rough thick ice with HH intensities between $-15$ dB and $-10$ dB.

6. Heavily deformed ice (HDefI): thick ice areas with very high degrees of deformation, thus having high HH intensities
($\geq -10$ dB).

For each class, 10 reference rectangles of $3\times3$ pixels are manually derived for each reference scene. Polygons of each class
in each scene are then randomly split in half to be used for training and validation. To improve the consistency of training across
scenes, polygons of LI, DefI and HDefI in a roughly $40\,km\times40\,km$ area surrounding the CO are derived for approximately the
'same ice' for all reference scenes, considering the shift in the position of the CO relative to scene borders. Fig. 3(a) shows
example reference polygons derived for the scene on 2019.11.22.

[Figure]

**Figure 2.** (a) TSX SC scenes in each month and OSI SAF sea ice classifications of surrounding sea ice areas in the middle of each month; (b) drift track of the weather station MetCity and its relative position to TSX SC scenes (c) 2m air temperature records through the study period and IA ranges of TSX scenes (average IAs in red line), with vertical lines representing selected reference scenes; (d) box-plots of NSIDC IST within each reference scene, where boxes cover the $25^{th}$ to the $75^{th}$ percentile with the median shown as the red bar. Whiskers extend to data extremes excluding outliers, and red crosses indicate outliers.

Fig. 4 shows an example of the progression of young ice on overlapping TSX SC and S1 EW scenes, both displaying the HH channel. High winds were observed during this period (Krumpen et al., 2021), which presumably contributed to large

135    ice opening and deformation events. One day prior to the example scenes (2019.11.20), wide-spread lead openings of open

[Figure]

[Figure]

**Figure 3.** (a) example reference polygons of different classes over the TSX SC scene on 2019.11.22; (b) dates and IA ranges of reference scenes. All subsequent figures of HH intensities use the dB range shown here.

water or nilas can be seen. Between 2019.11.20 and 2019.11.21, even more openings appeared which quickly re-froze into young ice. On the scene on 2019.11.21, most young ice appear very bright (thus belonging to the BYI class). While the leads gradually close up, these young ice areas gradually darken in TSX SC scenes until they reach a similar level of HH intensities to the surrounding ice. On the other hand, on S1 EW scenes, HH intensities for young ice gradually increase, from similar or

140    lower brightness to nearby LI to very bright on the 2019.11.23 and 2019.11.24, until they again reach similar brightness to LI later. This delayed increase and decrease in SAR intensities of young ice in C-band (5.405 GHz) compared to X-band (9.65 GHz) data is presumably due to different interactions between changing surface roughness scales and SAR signals of different wavelengths (Isleifson et al., 2010; Dierking, 2010; Barber et al., 2014; Park et al., 2020). This distinct difference in young ice intensities in the HH channel through time confirms the need for separating young ice into two classes for this study.

145    **2.3    IA dependency examination of HH intensities and textures**

**2.3.1    GLCM textures**

For the purpose of using image textures as features in sea ice classification, second-order texture measures (considering the relationship between groups of two pixels) are analyzed in this study, which are calculated on the basis of the gray-level co-occurrence matrix (GLCM, Haralick et al. 1973). The GLCM tabulates how different combinations of gray-levels co-

150    occur in pre-defined image windows, based on which statistical measures are derived to represent local spatial variations surrounding the central pixel. GLCM textures are among the most powerful texture discrimination tools (Barber and LeDrew,

[Figure]

[Figure]

**Figure 4.** Progression of young ice on coincident TSX SC and S1 EW scenes (both on the HH channel), all scaled by the same range of intensities through the 4-day period.

1991; Zakhvatkina et al., 2019), and have been widely used for texture-based classification of remote sensing images in general (Hall-Beyer, 2017), and specifically for sea ice classification of both X- and C-band SAR data (e.g., Clausi and Yu 2004; Leigh et al. 2014; Zakhvatkina et al. 2017; Murashkin et al. 2018; Park et al. 2020; Lohse et al. 2021; and those listed in Table 2).

155   Compared to classification based only on SAR backscatter intensities, they have been shown to provide additional separability between FYI and MYI, young ice and MYI, and level and deformed ice (e.g., Holmes et al. 1984; Shokr 1991; Leigh et al. 2014; Zakhvatkina et al. 2017; Lohse et al. 2021).

For TSX data, Ressel et al. (2015) used 5 GLCM textures calculated from the VV channel of 3 TSX SC images to classify sea ice near Svalbard with an artificial neural network (ANN), and reported satisfactory classification results for scenes with similar

160   IA ranges to the training scene. Liu et al. (2016) used 8 GLCM textures from TSX SC and Wide ScanSAR (WSC) as features to classify sea ice on the east coast of Antarctica, using IA directly as an input feature to a support vector machine (SVM) classifier. To reduce scalloping and inter-scan banding issues in ScanSAR images, Zhang et al. (2019) used a combination of Kalman filter, 5 GLCM textures and SVM on 5 TSX SC (HH/VV) scenes, and Liu et al. (2021) used the same 5 GLCM textures in two spatial scales from 8 TSX SC/WSC (HH) scenes to classify sea ice, both in the Beaufort Sea, with no corrections for

165   the IA effect. In this study, we examine the separability between sea ice classes provided by 17 commonly achievable GLCM textures (through ESA SNAP and GEE) calculated from TSX SC HH intensities, supported by the examination of their IA dependencies. This enables us to find an optimal way of using GLCM textures as input features into the GIA classifier, and classify sea ice for the MOSAiC drift with reliable correction of the IA effect. The GLCM textures used are listed in Table 1, where the mathematical expressions match those from Haralick et al. (1973) and Conners and Harlow (1980).

170   Texture windows surrounding the training pixels can potentially cover mixed classes. This is especially true for classes that are spatially confined, namely classes representing 'lead ice' (leads, DYI and BYI), and also HDefI. The former usually takes

[Figure]

[Figure]

**Table 1.** GLCM texture measures used analyzed in this study.

(1) Cluster Prominence (CLP):

$\sum_i \sum_j (i + j - \mu_i - \mu_j)^4 P_{i,j}$

(2) Cluster Shade (CLS):

$\sum_i \sum_j (i + j - \mu_i - \mu_j)^3 P_{i,j}$

(3) Contrast (CON): $\sum_i \sum_j P_{i,j} (i - j)^2$

(4) Correlation (COR): $\frac{\sum_i \sum_j ij P_{i,j} - \mu_x \mu_y}{\sigma_x \sigma_y}$

(5) Difference Entropy (DFE):

$-\sum_{i=0}^{N_g - 1} P_{x-y}(i) \log P_{x-y}(i)$

(6) Difference Variance (DFV):

$\sum_{i=2}^{2N_g} \left( i - \left[ \sum_{i=2}^{2N_g} i P_{x-y}(i) \right] \right)^2$

(7) Dissimilarity (DIS): $\sum_i \sum_j P_{i,j} |i - j|$

(8) Energy (ENG): $\sqrt{\sum_i \sum_j P_{i,j}{}^2}$

(9) Entropy (ENP): $\sum_i \sum_j P_{i,j} (-\ln P_{i,j})$

(10) Homogeneity (HOM): $\sum_i \sum_j \frac{P_{i,j}}{1+(i-j)^2}$

(11) Information Measure of Correlation 1 (IMC1):

$\frac{HXY - HXY1}{max(HX, HY)}$

(12) Information Measure of Correlation 2 (IMC2):

$\sqrt{(1 - \exp[-2.0(HXY2 - HXY)])}$

(13) Maximum Probability (MXP): $\max(P_{i,j})$

(14) Mean (MEAN): $\sum_i \sum_j i P_{i,j}$

(15) Sum Average (SMA): $\sum_{i=2}^{2N_g} i P_{x+y}(i)$

(16) Sum Variance (SMV):

$\sum_{i=2}^{2N_g} \left( i - \left[ \sum_{i=2}^{2N_g} i P_{x+y}(i) \right] \right)^2$

(17) Sum of Square: Variance (VAR):

$\sum_i \sum_j P_{i,j} (i - \mu)^2$

a linear shape along ice openings, and the latter usually includes linear strips or spatially limited aggregations of deformation features, or rounded MYI floes. Therefore, in the derivation of reference polygons, an effort was made to place polygons at the center of small, rounded features and along the width of linear features. Texture windows of mixed classes can still occur

175 for larger window sizes. Given these limitations, the maximum window sizes to guarantee the absence of mixed-class texture windows in the reference polygons in this study are 9, 25, 25, 81, 41 and 35 pixels (length of a window edge) for leads, DYI, BYI, LI, DefI and HDefI, respectively.

**2.3.2 IA dependencies**

In an initial examination of GLCM textures calculated from HH intensities in the linear and logarithmic (dB) domains, we

180 found that only textures of HH intensities in dB have a consistent linear relationship with IA, given properly constrained IA range (more details below). This is one of the pre-requisites for features to be used by the GIA classifier. Similar findings are reported in (Lohse et al., 2021) for C-band S1 EW data. Thus, GLCM textures are calculated for HH intensities in dB, split into 64 gray levels (to ensure balance between precision of gray-level information and computational efficiency) with a 2-pixel

offset averaged for 4 directions (0 , 45 , 90 , 135 ) to avoids directional sensitivity of textures. These texture parameters are
185 explained in Haralick et al. (1973), and in this study we use a data-driven approach to select the two most important texture
parameters for the purpose of image classification: texture window size and the combination of texture measures (details shown
in Section 2.3.3).

The distribution and scatter plots for HH intensities of the 13 reference scenes (IA range: of 17.18 to 59.56 , Fig. 5) show
that ambiguities in HH intensities are most prominent for two class pairs: BYI vs. HDefI and DYI vs. DefI. These difficult class
190 pairs are thus the focus of subsequent separability evaluations. The IA dependency of the leads class is the weakest among all
classes (Fig. 5), being mostly under the nominal noise floor (Fritz et al., 2013) and having the widest scatter in HH intensities.
HH intensities of other classes are relatively linear with IA through the whole IA range with significant slopes.

The distribution of GLCM textures calculated from dB intensities (in an example window size of 9 pixels), and their scatter
plots within the IA range of the 10 training scenes (IA: 31.90 to 59.56 ), are also shown in Fig. 5 (only the difficult class
195 pairs are shown for better visual clarity). Textures generally show a weak linear relationship with IA with varying levels of
dependencies (IA slopes), similar to previous C-band and X-band findings (e.g., Liu et al. 2016; Lohse et al. 2021; Scharien and
Nasonova 2020). Some textures show visually apparent separability between class distributions of one or both of the difficult
class pairs (e.g., DIS, ENP, MEAN, SMA, VAR) at this window size.

The classes form approximately Gaussian distributions both for HH intensities and most HH textures (Fig. 5), therefore
200 satisfying the pre-requisites to be used as features in the GIA classifier. A considerable part of the leads class is below the
nominal noise floor of the HH channel, affecting its distribution both for HH intensities and textures. Also, the leads class has
distinctly different HH intensities than other classes. Therefore, leads is not included in subsequent texture-based classification.
Instead, a separate classification is run for all scenes using HH intensities only, from which lead pixels are extracted and used
for the final classification result, which we found to provide satisfactory lead separation.

**2.3.3 Parameter optimization and calculation of GLCM textures**

Optimal combination of textures and window size are selected based on class separability. Window size is a key parameter in
texture-based image classification, and has been shown to influence classification accuracies more than other textural param-
eters (Marceau et al., 1990; Ferro and Warner, 2002). For a particular class pair, the optimal window size is dependent on the
spatial scales at which unique textures of the two classes can be statistically separated. Smaller window sizes fail to include the
210 characteristic variability of the classes, and thus prevent the separation between class distributions, while larger window sizes
reduce the effective resolution of the classification result, and introduce mixed-class texture windows. Similarly, the inclusion
of more texture measures generally adds to the classifier's ability to separate different classes, but the inclusion of redundant
texture measures leads to increased texture collinearity and longer computational time without improving classification ac-
curacy. The inclusion of a large number of texture measures and hence high dimensionality of the feature space additionally
215 leads to the risk of reduced classification accuracy (Hughes, 1968; Alonso et al., 2011). Therefore, it is desirable to minimize
the correlation between texture measures used for classification (Shokr, 1991; Hall-Beyer, 2008, 2017), particularly given our
purpose of classifying a winter time series. In summary, the main objectives in selecting optimal texture combination and win-

[Figure]

**Figure 5.** Histograms and scatter plots of TSX SC HH intensities with IA for the reference polygons (all classes), and GLCM textures (Table 1) with IA for the training polygons (only for DYI, DefI, BYI and HDefI for better visual clarity). Slope values of different class with IA are also shown (bold indicates statistical significance). Values of all texture measures are scaled to the -1 to 1 range to yield comparable slope values.

dow size are: 1. to provide separability between different classes, especially those having similar TSX SC HH intensities; 2. to keep a minimal window size to retain spatial details provided by the relatively high resolution of TSX SC data; 3. to reduce

220    correlation between textures used for classification.

[Figure]

Class separability is evaluated by the K-S distance (Massey Jr, 1951), which is non-parametric and thus a relatively robust metric without assumptions of class distribution (Daniel, 1990). The K-S distance quantifies the distance between class distributions, and the K-S test yields a test decision for the hypothesis that two classes come from the same distribution. The following procedures are used to derive an optimal windows size and texture combination:

225     1. for each texture, K-S distance values between class pairs are calculated for all window sizes between 3 and 61 pixels for all pixels within the training polygons (training pixels);

    2. for each combination of textures, the smallest window size at which all individual constituting textures provides statistical separability (evaluated by the K-S test) between all class pairs is selected as the 'optimal' window size;

    3. for each texture combination at its optimal window size, the summation of K-S distance values for all textures is divided 230 by the common logarithm of the summation of correlation coefficients between texture pairs within the combination, resulting in a 'combination rating' that provides control over texture collinearity. This rating is calculated for every texture combination from the 17 textures in Table 1;

    4. texture combinations with the 10 highest ratings (in corresponding optimal window sizes) are used to classify the training scenes, and these classification results are compared visually to arrive at a final selection of texture combination and window 235 size.

The selected optimal combination of textures is: COR, DIS, ENG, ENP, HOM, MAX and SMA. This texture combination has an optimal window size of 9 pixels. HH intensities and textures calculated accordingly for the training pixels are used to train the GIA classifier. Texture images are then calculated for all TSX SC scenes of interest and are used for final classification of the time series. GLCM texture calculation for the training pixels and the derivation of optimal texture parameters are conducted 240 using MATLAB 2021b (The Mathworks Inc., 2021). Further production of whole texture images using HH intensities are conducted using ESA SNAP and GEE.

**2.4   Classification of MOSAiC winter time series**

Sea ice classification is conducted on the time series using the GIA classifier trained with HH intensities and textures with the optimal texture combination and window size (shown above). Within the classification process, a Markov Random Field 245 (MRF) contextual smoothing component (Doulgeris, 2015) is added to alter the posterior class probabilities yielded from the GIA classifier before determining maximum probability class labels. This technique replaces global class probabilities with spatially varying local probabilities by giving more weight to class memberships of spatially neighboring classes. This process is added to reduce scattered mis-classified pixels created by the classification of texture images and also ScanSAR image artifacts including scalloping and inter-scan banding.

250     As the sea ice area surrounding the CO is the main focus of MOSAiC sea ice studies, in Section 3 we mainly present classification results for a 71 km×71 km square (hereafter referred to as subset A) and a 28 km×28 km square (subset B) surrounding the CO. For both subsets, a time series of areal fractions of each class is produced from the classification maps, providing a general assessment of relative changes of classes through the study period. Subset A provides results in a broad area surrounding the CO, while subset B serves to ensure a consistent sea ice area for class fraction calculation, as significant

255 parts of subset A are often outside of TSX SC scene boundaries, resulting in the inclusion of a considerable amount of different ice surfaces when calculating class fractions.

**3 Results and discussion**

**3.1 IA dependencies and GLCM textures - this and previous studies**

IA slopes of C-band and X-band SAR backscatter intensities for different sea ice types derived in previous studies are shown

260 in Fig. 6. Different classification schemes are used for these studies, and we summarize them into 5 commonly used classes, and display per-class IA slopes for their classes that are the most closely related to these 5 classes. There are limited number of studies reporting IA dependencies of Arctic sea ice types for X-band sensors. IA slope values shown in Liu et al. (2016), presented in blue asterisks, are derived from TSX SC and WSC scenes with a limited IA range of 22.61 to 45.31 from the east coast of Antarctica. HH intensities of TSX SC data derived in this study are generally less dependent on IA than those for

265 C-band sensors (values summarized in Guo et al. (2022)), which is also observed in previous comparative studies of airborne X- and C-band sensors (e.g., Mäkynen and Hallikainen 2004). The general pattern of comparative IA slopes between classes is similar for C- and X-band: LI has a slightly stronger IA dependency than deformed FYI and MYI (in this study DefI and HDefI), presumably due to stronger volume scattering and added randomness in backscatter caused by deformation features for these two classes, both leading to decreased sensitivity to IA (Mäkynen et al., 2002; Dierking and Dall, 2007; Zakhvatkina

270 et al., 2013). These differences confirm the necessity of per-class IA correction in classifying the time series.

[Figure]

**Figure 6.** Comparison between IA slope values derived in this and previous studies. Dots are for C-band results, and asterisks for X-band ones. Correspondence between ice classes shown in the figure and closest ice classes in the original studies that are defined differently or more specifically: [1] FYI: FYI with dry snow on top; [2] FYI: land-fast smooth FYI with thin (7.7±3.9 cm) to thick (36.4±12.3 cm) snow cover; [3] Leads: nilas; YI: deformed gray ice; [4] MYI: averaged for MYI and old MYI; [5] YI: averaged for DYI and BYI; Deformed FYI: HDefI; MYI: DefI.

[Figure]

[Figure]

Table 2 shows GLCM texture parameters used in previous sea ice classification studies using X-band SAR. The table shows a wide variety of datasets, texture combinations and window sizes (in terms of physical length of window boundaries in meters), indicating that various GLCM textures on different geographical scales are useful for discriminating between sea ice classes. This is given that many studies use a limited number of textures measures and do not involve a process of selecting texture

275 combinations based on class separability and texture collinearity. CON, COR, ENP and HOM are examples of frequently used textures in sea ice classification. The texture parameter selection workflow established in this study produces satisfactory classification results (Section 3.2 and Section 3.3) and is generally applicable for future studies. However, the optimal texture window size and combination yielded in this study is specific to the dataset (on the constrained IA range of the training scenes) under examination, with an aim to provide separability between classes while minimizing spatial smoothing due to texture

280 calculation.

**Table 2.** Texture parameter selection in this and previous studies.

| | Data | | | | Texture parameters | |
|---|---|---|---|---|---|---|
| | **Area** | **Dataset** | **Frequency & channel**[1] | **Resolution**[2] **(m)** | **GLCM textures**[3] | **Window size**[3] **- pixel (m)** |
| Holmes et al. (1984) | Beaufort Sea | SURSAT SAR-580 (airborne) | X-band HV | 3 | CON, ENP | 5 (15) |
| Barder & LeDrew (1991) | Mould Bay, Canada | STAR-1 (airborne) | X-band HH | 6 | UNI[4], COR, ENP, DIS, CON | 25 (150) |
| Shokr (1991) | Mould Bay, Canada | STAR-1 (airborne) | X-band HH | 36 | CON, ENP, UNI[4], HOM, MAX | 5 (180) |
| Liu et al. (2016) | East coast, Antarctica | TSX SC/WSC | X-band HH | 15 | ASM, CON, COR, DIS, ENP, HOM, MEAN, VAR | 39 (585) |
| Ressel et al. (2015) | Barents Sea | TSX SC | X-band VV | ~48 | CON, DIS, ENG, ENP, HOM | 11 (~528) |
| Zhang et al. (2019) | Barents Sea | TSX SC | X-band HH/VV | 8.25 | CON, COR, HOM, MEAN, VAR | 39 (321.75) |
| Liu et al. (2021) | Beaufort Sea | TSX SC/WSC | X-band HH | 8.25 | CON, COR, HOM, MEAN, VAR | 39 (321.75) |
| This study | MOSAiC Drift | TSX SC | X-band HH | 8.25 | COR, DIS, ENG, ENP, HOM, MAX, SMA | 9 (74.25) |

[1] Only SAR channels used for GLCM calculation are shown.
[2] Effective pixel spacing after pre-processing.
[3] GLCM textures and window sizes are those used for final classification.
[4] UNI: Uniformity $= \sum_i \sum_j \mathsf{P}_{i,j}^2$, therefore similar to ENG.

**3.2 Classification with HH intensities and textures**

Classification results for three example scene subsets across the time series are shown in Fig. 7, where classification using HH intensities only, HH intensities and textures, and with additional application of MRF contextual smoothing are compared. Considerable classification improvement can be seen from the inclusion of GLCM textures, especially in the correct separation

285  between the difficult class pairs. Due to ambiguities in HH intensities, classification without textures shows prevalent mixing of those class pairs. DefI and HDefI are frequently mis-classified as young ice (e.g., 2020.01.08 and 2020.03.28, zoomed-in image patches), resulting in classification maps dominated by DYI and BYI (green). Young ice is also frequently classified as DefI or HDefI (e.g., 2019.11.14, zoomed-in image patch). This issue is largely remedied by the inclusion of textures into the classification. MRF contextual smoothing further fulfills the intended purpose of eliminating scattered mis-classified pix-

290  els due to texture calculation and image artifacts, which are small in areal coverage but wide-spread, thus necessitating a smoothing process. The average overall accuracy calculated from the training scenes for classification on HH intensities and textures (82.09%) is significantly higher (p-value $< 00$) than that on HH intensities only (69.75%). The use of MRF contextual smoothing further increases (p-value $< 00$) the overall accuracy to 86.05%. The MRF contextual smoothing technique is theoretically (Doulgeris, 2015) and practically (not shown) superior to image smoothing processes without the consideration

295  of contextual information, e.g., a local majority filter, in improving classification accuracy and minimizing the loss of spatial detail.

**3.3  Winter sea ice classification time series surrounding the MOSAiC ice camp**

**3.3.1  Classification maps**

Resulting classification maps in subset A in the middle of each month, as well as the last scene of the time series (2020.03.28),

300  are shown in Fig. 8. The general distribution of LI vs. DefI and HDefI is consistent through the time series in the areal extent of subset A, as well as for the MOSAiC ice floe carrying the CO (zoomed-in patches). For reference, a manually derived classification of a small area around Polarstern produced by an co-author with extensive knowledge of sea ice conditions in MOSAiC is shown in Fig. 9(a). Our classification is consistent with ground observations (summarized in the manual classification map) indicating that the MOSAiC ice floe was composed of a mixture of FYI and SYI, with a strongly deformed zone

305  in the center named 'the Fortress' (the oval-shaped ice surface classified consistently as DefI and HDefI, i.e., dark red or light red) (Krumpen et al., 2020; Itkin et al., in review). In most scenes in November 2019, part of the SYI surface in the MOSAiC ice floe surrounding the Fortress appear similar to or even darker than nearby LI (Fig. 8), thus classified as LI. This has been observed to be attributable to the presence of re-frozen melt ponds (Fig. 9(a); Krumpen et al. 2021). The classification maps clearly capture the break up and change of size and shape for the MOSAiC ice floe. Major lead openings are seen on 2020.03.17

310  and 2020.03.28, which are identified as BYI and DYI. Panoramic photos taken from Polarstern (Fig. 9(b), Marcel et al. 2021) confirm the presence of ice openings occupied by young ice with the same relative positioning to the ship as indicated by Fig. 8 (Polarstern circled in black in the zoomed-in patches).

  The standard deviation of sea ice thickness measured from the electromagnetic induction (EM) instrument (GEM-2, Hendricks et al. 2022) along several transects near the CO are used as a combined indicator of sea ice surface and bottom roughness,

315  and is plotted in blue on the classification maps in Fig. 10(a). Manual correction of sea ice roughness data point positions is conducted to account for sea ice drift and and geo-location errors of the sensor. The effect of ice floe rotation and deformation is still present, and the data points are averaged for windows of $4 \times 4$ TSX SC pixels (thus $3\partial \times 3\partial$) to partially remedy

[Figure]

**Figure 7.** Example TSX SC scenes and classification maps using the GIA classifier trained with HH intensities only, HH intensities and the optimal texture measures, and additionally with MRF contextual smoothing applied, for 3 dates across the time series. All HH subsets are scaled by the same range of intensities, with IAs shown as contours.

these issues. It can be seen that rougher ice (deeper blue) along the transects mostly correctly correspond to areas classified as DefI or HDefI, and smoother ice (lighter blue) to LI (Fig. 10(a)).

320    Additionally, a classification is conducted on sea ice roughness transects (into LI, DefI and HDefI) for comparison with the SAR-based classification (details of the method shown in Itkin et al. in review), which is shown in Fig. 10(a) on top of HH intensities in the same color scale as the SAR-based classification. In areas of mostly smooth FYI and SYI (outside the Fortress), sea ice roughness is classified with a threshold of 0.2 m into LI (yellow) and DefI (dark red), corresponding well with LI and DefI in the SAR-based classification result. Inside the Fortress, sea ice roughness is classified using the same

325    threshold into DefI (dark red) and heavily HDefI (light red), again showing similar spatial distribution to DefI and HDefI on the SAR-based classification maps.

[Figure]

**Figure 8.** Classification maps of TSX SC scenes near the middle of each month in the time series, and zoomed-in subsets focusing on the MOSAiC ice floe. The last scene in the time series is also shown. On the zoomed-in classification maps, black circles indicate the position of Polarstern. All HH scenes are scaled by the same range of intensities.

[Figure]

[Figure]

**Figure 9.** (a) Manual sea ice classification of the CO overlaid on a RS2 SCW scene (HH) on 2020.01.12 - yellow: FYI; purple: rough SYI; red: ponded SYI; Polarstern, weather stations, and transects with sea ice thickness measurements are also shown; (b) Panoramic photos taken from Polarstern on 2020.03.17 and 2020.03.28.

Two repeated transects are available on all dates in Fig. 10(a): the Southern and Northern transect loops, or 'Sloop' and 'Nloop'. Sloop is located in the aforementioned ponded SYI area, and crosses a mixture of rough SYI and smooth refrozen melt pond surface which have similar X-band backscatter to LI, while Nloop is located within the Fortress and thus is a transect 330 of predominantly heavily deformed SYI (Itkin et al., in review). These observations are mostly correctly shown both in our classification map and the classified transects. The transect 'runway', established on a LI surface to supplement FYI sampling in the CO, is also consistently classified as LI in both classifications in Fig. 10(a).

As sea ice roughness calculated from ice thickness represents both surface and bottom roughness, apparent mismatch between the two classifications can be seen in the transect Sloop on the 'November ridges,' most notably on 2020.01.12, as 335 pointed out by arrows in Fig. 10(a). In the southern part of Sloop (arrow to the right) where ice is relatively smooth but thick, high roughness is calculated from the transect (classified as DefI), while TSX SC HH intensities are low (thus classified as LI), presumably due to the dominance of ice-bottom roughness. On the contrary, in the northern part of Sloop (arrow to the left), high HH intensities are observed indicating rough ice surface (classified as DefI), while ice roughness calculated from the transect is low and mostly classified as LI, most likely due to low standard deviations calculated from the relatively thin ice 340 (Fig. 9(a)) in this area.

The percentages of correct correspondence between this *in situ* ice roughness classification and the SAR-based classification for repeated transects through the time series is shown in Fig. 10(b). Corresponding to a specific *in situ* ice roughness classi-

[Figure]

**Figure 10.** Sea ice roughness in transects (in blue) overlaid on classification maps, and classification of sea ice roughness (in the same color scheme as the SAR-based classification) overlaid on TSX SC HH intensities, in several transects within a 4 km×4 km square around Polarstern for the same dates as Fig. 8; (b) percentage of correspondence between SAR-based and sea ice thickness-based classification of ice roughness in repeated transects in the CO; (c) scatter plot of HH intensities vs. sea ice roughness in repeated transects in the CO, grouped by their corresponding class labels from the SAR-based classification.

[Figure]

fication data point, the SAR-based ice classification is counted as 'correct' if the ice class of at least one TSX SC pixel in its surrounding $4 \times 4$ pixel window reports the same class. Good correspondence is found between the two types of classification,

345 with the percentages of 'correctly' classified SAR pixels being consistently near or over 80%.

Finally, we demonstrate the relationship between HH intensities and sea ice roughness for data points in repeated transects in Fig. 10(c), grouped by class labels from the SAR-based classification result. An apparent logarithmic fit can be seen, where the points representing mean roughness and intensities for the 3 thick ice classes (LI, DefI and HDefI, shown in stars) are very close to the fitted curve. This indicates that TSX SC HH intensities are controlled largely by sea ice roughness for the training

350 scenes, which, together with the good correspondence between the SAR-based and *in situ* ice roughness classifications, justifies our chosen classification scheme which separates thick ice into different degrees of deformation. Previous studies have also found winter sea ice roughness to be the dominant factor of C- and L-band SAR intensities for specific channels (e.g., Dierking and Dall 2007; Gegiuc et al. 2018; Cafarella et al. 2019; Segal et al. 2020). As mentioned before, both surface and bottom roughness contribute to the variation in the calculated sea ice roughness, presumably contributing to the relatively wide spread

355 of the scatter plot.

**3.3.2 Temporal development of ice class fractions**

Areal fractions of different classes in subsets A (solid lines) and B (dashed lines) for all scenes in the time series are shown in Fig. 11(a) and (b). Leads, DYI and BYI are combined into a 'lead ice' category, representing areas of ice opening. The lead ice fraction of a certain scene depends on where the ice openings are within their 'life cycles,' i.e., typically from open water to

360 nilas, young ice with various levels of SAR backscatter intensities, and eventually FYI (in our classification scheme from leads to DYI, BYI, DYI and then LI). For example, a major divergence event occurred from 2020.02.26 to 2020.02.29, resulting in a long ice opening on 2020.02.29 in subset A (not shown) which is not registered in Fig. 11(b) due to its similar intensities and textures to LI.

Relative proportions of LI vs. DefI and HDefI are reasonably consistent through the time series (Fig. 11(a)). Lead ice

365 fractions in subset A are generally higher than subset B, indicating consistently more ice openings captured in subset A, which is considerably larger than subset B (Fig. 11(b)). Several peaks of lead ice fraction are visible through the time series, most notably in mid-late November and mid-December 2019, early January and early and late February 2020. Starting from early March 2020, lead ice fractions remain high throughout the month for both subsets. During this period, ice openings can be consistently visually observed in the scenes. A major ice opening event occurred on 2020.03.28 (Fig. 8), where lead ice fraction

370 reached 4.98% for subset A. This event persisted through early April. A more detailed examination of the ice opening events is conducted by comparing the class fractions to indicators of ice openings from *in situ* data derived in this and other MOSAiC studies:

1. Areal change between buoys (Fig. 11(b), orange): the area between 4 selected buoys (P103, P193, P195 and P204) surrounding the CO (Bliss et al., 2021) are calculated every 3 hours, partially representing events of divergence and convergence.

375 Similar peaks in this areal change can be seen in mid-late November to those of the lead ice fractions (Fig. 11(b)). The area also exhibit frequent changes through March, indicating frequent short-lived ice openings between the buoys. The sharp decrease

[Figure]

[Figure]

**Figure 11.** Fractions of (a) FYI and MYI, and (b) lead ice, over the time series. In (a) and (b), solid lines represent class fractions derived from subset A, while dashed lines represent those from subset B. Areas between 4 buoys surrounding the CO (names and positions shown for 2019.11.01) over the time series are also plotted in (b).

in the area starting from 2019.12.15, caused by a large-scale shearing event across the entire extent of subset A lasting through 2019.12.23, leads to mildly increased lead ice fractions in mid-December (Fig. 11(b)). The area between the buoys remain relatively constant in other periods in the time series.

380    2. Other MOSAiC studies: several peaks of lead ice fractions have good correspondence to lead fractions generated from optical satellite observation (Reiser et al. 2020, reported in Krumpen et al. (2021)). Lead fractions within a 50 km radius from the CO show prominent peaks in early-mid December, early February and early and late March, matching those given in Fig. 11(b). No usable lead fraction is produced for mid-late November and mid-late March, but prominent divergence-convergence events can be seen in mid-late November and late March as obtained from S1 sea ice drift data (Krumpen et al.,

385    2021). In a recent study of sea ice classification on TSX dual-pol StripMap images (SM, 54 km×16 km at 3.5m resolution) using convolutional neural networks and a conditional random field, Kortum et al. (2022) also identifies prominent rise in young ice class fractions in the 3 km×3 km area around Polarstern in late November 2019 and late March 2020. Abrupt and prominent changes in wind speed and direction were recorded from Polarstern in mid-late November 2019 and mid- to late-March 2020 (Itkin et al., in review), which likely contributed to lead opening events shown in Fig. 11(b). These comparisons

390    demonstrate that the classified time series is valuable as an indicator of ice openings, and thus a good reference to studying associated physical processes through the expedition in a larger spatial scale than the previously derived MOSAiC sea ice classification product (Kortum et al., 2022).

[Figure]

**3.4 Limitations and future steps**

The current classifier has limited capability in detecting linear young ice areas that are narrower than the texture window
395  size used. This is an inevitable outcome of texture-based classification, which contributes to our objective to minimize texture
windows. The leads class are mostly fully represented in the classification maps, as it is classified with HH intensities only.
The inherent scalloping and inter-scan banding issues for ScanSAR scenes can be seen in the TSX SC scenes, and is more
prominently shown in the texture images. These issues are more pronounced in scenes with IAs of higher than 50 , and affects
the classification results, most notably leading to mis-classification between the difficult class pairs. In this study, this issue is
400  partially remedied by the MRF contextual smoothing process. If future applications necessitate the usage of TSX SC scenes
with obvious sensor artifacts, additional steps should be taken to remedy these issues using procedures proposed by previous
studies (e.g., Iqbal et al. 2012; Yang et al. 2020), as used in e.g., Zhang et al. (2019). Also, no continuous *in situ* observation of
thin ice is available to provide detailed information on the evolution of these areas through the time series. Sea ice roughness
derived from *in situ* sea ice thickness measurements is calculated on a different spatial scale than the classification maps,
405  represents both surface and bottom ice roughness, and suffers from potential co-location errors due to sea ice rotation and
deformation. Therefore, the utilization of ice surface roughness data calculated from air-borne and ground-based laser scanners
is desirable in future studies as a stronger validation of ice classification.

This study has focused on the freezing season during the MOSAiC expedition. Reasonable classification of April scenes are
achievable through individually fitted classifiers with different texture parameters, but a consistent classifier is not available due
410  to the small IA range of April scenes for which separate training is needed in the framework of the proposed workflow. Future
steps will extend the study period into the summer season to examine the seasonality of TSX SC textures on sea ice, and its
effects on texture-based sea ice classification. This seasonality analysis is valuable in providing a comprehensive perspective of
the change in IA dependencies of TSX SC textures from freezing to melting. Further application of the proposed classification
workflow can be extended to dual-polarization TSX SC data, utilizing information from the additional VV channel. Similarly,
415  studies of C-band SAR-based sea ice classification have demonstrated that per-class IA correction is still necessary for the HV
channel, despite the weaker IA dependencies of its intensities compared to HH due to much more pronounced influence from
noise (Aldenhoff et al., 2020; Lohse et al., 2021).

For future studies on texture-based sea ice classification, more detailed quantification of the correspondence between GLCM
textures and sea ice surface properties should be conducted, following previous studies (e.g., Baraldi and Parmiggiani 1995; Soh
420  and Tsatsoulis 1999). Also, as mentioned earlier, previous studies of SAR texture-based sea ice classification has achieved sea
ice type separation in various physical window sizes. Therefore, further investigation into better including multi-scale textural
information (by varying window and displacement step sizes) is desirable, which can potentially capture and utilize both micro-
and macro-scale sea ice textures (e.g., Soh and Tsatsoulis 1999; Leigh et al. 2014). Finally, although GLCM textures have been
demonstrated to be among the most powerful tools for texture-based classification (Hall-Beyer, 2017; Zakhvatkina et al., 2019),
425  it is still valuable to investigate IA dependencies and utilization of other types of image textures previously used for sea ice
classification, e.g., first-order textures, image moments, MRF-based, wavelet transformed-based, variogram-based, and Gray

[Figure]

Level Dependence Matrix (GLDM) based textures, etc. (Conners and Harlow, 1980; Unser, 1995; Clausi, 2001; Clausi and Yu, 2004; Sanden and Hoekman, 2005; Bogdanov et al., 2007; Komarov and Buehner, 2017; Gegiuc et al., 2018; Scharien and Nasonova, 2020).

**430    4   Conclusions**

This study demonstrates per-class IA dependencies of HH intensities and GLCM textures calculated from TSX SC data, based on scenes acquired during the winter months of the MOSAiC expedition. Linear IA dependencies of HH intensities in dB are shown to be generally lower than C-band data, but between-class IA slope differences still necessitate per-class correction of the IA effect. In constrained IA ranges, GLCM textures calculated from dB intensities also exhibit linear dependency to IA, 435 and is thus suitable for use by the GIA classifier. The leads class has a relatively wide scatter in HH intensities and textures vs. IAs resulting in weak linear dependency, and is thus retrieved from a separate classification based on HH intensities only. A texture parameter selection process based on statistical separability between class distributions has determined the optimal texture combination to be used is COR, DIS, ENG, ENP, HOM, MAX and SMA (see Table 1 for definitions) at a window size of 9 pixels. With a classification scheme designed for the task of one-band classification, the GIA classifier is trained 440 using HH intensities and textures to account for different IA dependencies of different ice classes, and used to classify the time series. Qualitative (through visual inspection of resulting classification maps) and quantitative (using classification accuracies calculated from validation polygons) assessments show that the inclusion of GLCM textures brings vast improvements in classification performance compared to HH intensity-based classification, and is essential in classifying TSX SC data. The application of MRF contextual smoothing further refines the classification result while preserving maximum spatial details, 445 leading to significantly increased classification accuracies. Good correspondence is found between the classification result and sea ice roughness calculated from *in situ* sea ice thickness measurements. The classified time series show reasonably consistent fractions of LI vs. DefI and HDefI through the time series. Lead ice fractions derived from the classification result correspond well with other indicators of ice openings derived in this and previous studies. This suggests that the classified time series can serve as a reliable reference of the changing sea ice conditions and associated physical processes during the expedition within 450 the spatial scale of TSX SC scenes (approximately $100 \times 150$ km). This study provides valuable information on the utilization of per-class IA dependencies of TSX SC intensities and GLCM textures in classifying sea ice, and a classification product of a broad area surrounding the MOSAiC ice camp that can potentially facilitate future MOSAiC sea ice studies and modeling efforts.

*Data availability.*   The classified time series in subset A is available as projected GeoTIFFs (in EPSG:3575) here: https://www.dropbox.com/
455 sh/edx4eq2oux0fqdg/AAB5CXZ8ReTwZNpXe48mpoZYa?dl=0. Correspondence between pixel values and class labels: 3: leads; 5: DYI; 6: BYI; 7: LI; 9: DefI; 10: HDefI.

[Figure]

*Author contributions.* PI, SSI, APD, MJ, and GS were involved in project administration and supervision. All co-authors were involved in the conceptualization of the study. SSI was responsible for TSX SC data acquisition. WG was responsible for data curation, methodology designing, formal analysis, and result visualization. PI conducted the analysis on sea ice surface roughness using sea ice thickness measure-

460    ments from MOSAiC, and provided the manual sea ice classification map surrounding the CO and the time series of areal change between the buoys. APD provided his codes and knowledge of MRF contextual smoothing. WG prepared the manuscript, with contributions from all co-authors in reviewing and editing.

*Competing interests.* No competing interests are present.

*Acknowledgements.* The authors would like to thank all persons involved in the expedition of the Research Vessel Polarstern during Multi-

465    disciplinary Drifting Observatory for the Study of the Arctic Climate (MOSAiC) in 2019–2020 as listed in Nixdorf et al. (2021). This work was supported by the Research Council of Norway (RCN) projects: Sea Ice Deformation and Snow for an Arctic in Transition (SIDRiFT) (287871), Center for Integrated Remote Sensing and Forecasting for Arctic Operations (CIRFA) (237906), and Project Oil spill and newly formed sea ice detection, characterization, and mapping in the Barents Sea using remote sensing by SAR (OIBSAR) (280616). SSI and GS are supported by the Deutsche Forschungsgemeinschaft (DFG) through the Project "MOSAiCmicrowaveRS" (Grant 420499875).

470    Data used in this article were produced as part of the International Multidisciplinary Drifting Observatory for the Study of the Arctic Climate (MOSAiC) with the tag MOSAiC20192020 and Project_ID: AWI_PS122_00. TerraSAR-X images used in this study were acquired using the TerraSAR-X AO OCE3562_4 (PI: SS). RADARSAT-2 data was provided by NSC/KSAT under the Norwegian-Canadian Radarsat agreement 2019 and 2020. Sentinel-1 data is publicly available from the Copernicus Open Access Hub (https://scihub.copernicus.eu/, last access: Oct 2021; European Space Agency 2021). The OSI SAF global sea ice type product (OSI-403-d) is publicly available from https:

475    //osi-saf.eumetsat.int/products/osi-403-d (last access: Oct 2021; OSI SAF 2019). The NSIDC IST dataset (MOD29/MYD29) is publicly available from https://nsidc.org/data/MOD29 (last access: Oct 2021; Hall and Riggs. 2021).

[revised manuscript text omitted]

---

## Author Response (AR1)

**Reviewer 1**

The manuscript "Sea ice classification of TerraSAR-X ScanSAR images for theMOSAiC expedition incorporating per-class incidence angle dependency of image texture" presents methodology and results of sea ice type classification of TerraSAR-X imagery obtained during the MOSAiC expedition. Despite very interesting findings, due to large diversity of methods, results and analysis and a large size of the manuscript, it is recommended to split the manuscript in two parts, improve the order of the presentation and resubmit the manuscript(s) after a major revision.

**Major comments**

1. Although only two objectives are formulated in the introduction, the impression is that the manuscript attempts to fulfil at least four: 1. Investigate per-class AI dependence; 2. Optimize parameters of texture features; 3. Train and evaluate classifier; 4. Analyse time series. In my opinion such variety of objectives does not allow to focus well. That makes the manuscript too long to read and the story too difficult to follow. I would suggest to completely remove section 3.3 and correspondingly reduce section 3.4. I'm confident that results shown in these sections deserve a separate paper. I will therefore focus my review on the first, methodological part.

We think it's reasonable to maintain the current manuscript structure, while streamlining the text for better clarity and readability, sharpening the aim of the manuscript to 2 points: introducing a classification product, and demonstrating TSX IA dependency in support of the classification.

As suggested by the title, this paper mainly serves as an introduction to a classified time series which can be a useful dataset for MOSAiC-related research. Quantitative evaluation of the classification, the comparison to ice roughness transects, and the comparison to ice opening records in other studies have shown the reliability of the classified time series.

Method development was not central, having used an established classifier and commonly used GLCM textures to aid the classification. The parameter optimization process is implementation, rather than development. The demonstration of IA dependencies of TSX intensities/textures, which is another major finding of the paper, secondary to the main objective, is also based on the time series itself.

The manuscript has been re-aimed around these 2 points in all sections, with more emphasis placed on demonstrating the IA dependencies & relating the classification results to the MOSAiC mission. The text is more concise and the length of the methodology part has been reduced.

2. What is GIA classifier? Authors refer to that term in many places, but it is never defined or explained. I guess, that's one of the central blocks in the classification algorithm: apparently, the backscatter, the texture features, the IA are passed into the mysterious "GIA classifier" for doing the actual classification. But how?! I'm very curious to know. GIA classifier needs to be clearly explained.

As stated in the text, the GIA classifier was developed and published in Lohse et al., 2020. The investigation of IA dependencies of sea ice types on TSX SC is central to this study, hence the choice of the GIA classifier which specifically incorporates this phenomenon well. This is explained by the most part of the 3rd paragraph in the Introduction. We think that this length is suitable for the manuscript which is currently already lengthy.

3. Order of presentation needs to be revised in order to correspond to the selected logic (Intro, Data and methods, Results and Discussion): Lines 133 - 144 and Fig. 4 should come in Section 2.1 Data; Lines 151 - 164 with Table 1 and lines 271 - 275 with Table 2 belong to Introduction as they describe state-of-the-art; Section 2.3.2 belongs to Results as it describes WHAT is achieved and not HOW it is achieved.

**Edited.**

**Exceptions:**

a. Table 1 is specific to this study. Therefore, we think it's best to keep it in its current position.

b. Section 2.3.2 demonstrates IA dependencies, which we think is secondary to the main aim of introducing the classification product, and belongs to the 'Materials' category, introducing one of the characteristics of the dataset. We wish to focus the Results & Discussion part to evaluating the product and relating it to other MOSAiC products/studies. Therefore, we think it's best to keep Section 2.3.2 in the M&M section.

4. Analysis of IA dependence for various ice types need to be increased as it is an important result of this work. What is error-bars of the slopes (it can be computed, e.g. by bootstrapping) and what is significance? What is the reason for large positive bias of the slopes – speculation on stronger volume scattering needs to be expanded. What is physical reasoning behind positive slope for leads? The suggested method and parametrisations seem to be difficult to use in other conditions (C-band, other IA, other ice types, summer). Although it is mentioned as a limitation in the end, I believe it is important to also underline in the Introduction – the goal is to study a specific TSX SC timeseries and for analysis of another dataset a similar full-scale analysis needs to be performed.

In the current fig.4, IA slope values in bold fonts indicate statical significance of the linear regression model, while regular fonts indicate otherwise. For HH intensities, all slope values are significant except for the leads class, which is expected as all leads pixels are under the noise floor of the sensor, leading to a wide distribution of pixel values that does not exhibit a significant linear dependency to IA (mentioned in the text). The positive slope is therefore not significant and has no physical meaning (visually, a slight negative slope can be observed). Errors can be computed for the linear models but is of less interest to our study, but can be included in the appendix if desired.

Yes, an explanation is added to the introduction to clarify the limited setting of this study: 'In summary, the objectives of this study are: 1. to investigate and demonstrate per-class IA dependencies of TSX SC HH intensity and GLCM textures **specifically for the above mentioned study area and period**....'

5. Image size and number of texture features are undoubtfully important hyperparameters of the Haralick algorithm. However, neglecting quantisation level and distance to neighbour pixels can lead to significantly worse results. Sensitivity to these two parameters should also be studied, for example in this respect: how does despeckling boxcar filter impact the GLCM? In theory, if a 3x3filter is applied and then GLCM is computed with 2 pix distance, there should be almost no elements in GLCM off the main diagonal. On another note, Haralick (1973) suggested using adjacent pixels (d=1) so the choice of authors d=2 should be tested and explained better.

The number of quantization levels directly impacts the precision of the converted integer values used for GLCM calculation in representing the actual pixel values. Therefore, within reasonable computational loads, more levels are desirable. A level of 64 is thus chosen, and the reasoning is included in the text.

The speckle filter is no longer used to further preserve spatial details.

The displacement value is now directly added to the parameter optimization process (section 2.3.3), i.e., an optimal set of displacement size, window size and texture combination is selected together.

Minor comments

L7. Phrases in parenthesis make the sentence very unclear. Please split into two sentences.

All instances in manuscript are adjusted accordingly, except for very short clarifications and acronyms.

L12. Unfortunately the GIA classifier and class probabilities are never explained in the manuscript.

See reply to major comment 2.

L24. Please provide reference to prove the "largest expedition in history"?

Reference added.

L71. Objective 1 is actually two objectives: 1 . to investigate and demonstrate per-class IA dependencies of TSX SC HH intensity and GLCM textures; 2. to determine the feasibility and optimal parameterization of including texture measures as input features to the GIA classifier.

This paragraph has been re-edited to more clearly show our main aims – introducing the classification product, while demonstrating IA dependencies.

Figure 1 shall be removed as it does not explain anything. Edited.

L96. "and shown in details in " -> ", dates shown in" Edited.

L107. Why were these ice categories chosen? It should be written that other categories were not present in the dataset and the method cannot be extrapolated.

'Based on the ice conditions in the study area and period' is added to show that the choice is made considering the specific conditions of this study.

L129. Polygons == rectangles? This is unclear.

The first instance has been edited to 'polygons in rectangles,' and 'polygons' are used subsequently.

L133. Maybe "evolution of young ice" ? Edited.

L135. Please rewrite "wide-spread lead openings of open water or nilas" as "wide opening of leads with water or nilas"

This is meant to emphasize the spatial abundance of leads within the area, instead of the physical widths of the leads. It has been edited to ' wide-spread opening of leads with water or nilas.'

Figure 3. The smallest sub-images seem to be very blurred. Is it the effect of the despeckling filter or just visualisation?

The smallest sub-images are zoomed in to a level where individual pixels are visible, in order to give a (example) visual impression of textures of each class at this scale. This visual effect is natural at this zoom scale, and also it can be seen that different classes have different 'blurriness' which are related to how their textures differ from each other. In the current revision, we no longer apply a speckle filter on the images, and visually these subsets are now less 'blurred,' as expected.

L208. Cannot agree here. Other authors also studied distance and number of grey levels (e.g. Clausi 2002). Sensitivity to these two parameters need also to be studied (see major comments). See reply to the corresponding major comment.

L221. "...and thus is a relatively..." Edited.

L236 and 237. Is that already results of parameter optimization? Then it is better to move to the Results section. Edited.

L268. Whay volume scattering is presumed to be stronger?

In this sentence, 'stronger volume scattering' refers to MYI, and 'added randomness in backscatter caused by deformation features' refers to deformed FYI. 'Respectively' is now added at the end of the sentence to avoid confusion.

Figure 6. Is positive slope for leads even physical? How the strong positive bias of the slopes can be explained?

All of these pixels are under the noise floor of the sensor, resulting in unreliable IA dependencies. Explanation is added to the text.

L274. "This is given that" can be removed. Edited.

L276 – 280. This seems to be logical after the results, in the Conclusions section. As this relates to the limitations of this study, these sentences have been moved to the 'Limitations and future steps' section, which also avoids lengthy conclusions.

L321. A reference to unpublished work just supports my concern that it is too early to include this section in the manuscript.

This sentence only refers to the method of classifying the sea ice roughness transects that happens to be also used in another study. These roughness transects are analyzed specifically for this study.

Figure 10. It is impossible to see shades of blue on the roughness transects. The symbology is adjusted for better visualization with thicker transects.

Figure 11 and Lines 389 - 392. Why the 10% sudden drop of the polygon area on ~15 December is not reflected in a similar change of MY or young ice? Why does the lead ice increases ~3 times on 1 March and this is not reflected in the polygon area? Where are the plots of "other mosaic studies" that are easy to compare with fractions of different ice types? I'm afraid it is too early to write "that the classified time series is valuable as indicator of ice openings" as I cannot see a proof of that. Instead, the variations of ice fractions seem to be rather spontaneous and connected to uncertainty of the algorithm. As mentioned in the text, the polygon formed by the buoys is a small, variable area around the ship, which is much smaller than the parts of TSX scenes used for classification. Therefore, not all variations are synchronized between these time series. We'll consider putting results from other papers directly within the plots for easier comparison.

L396. "The leads class are mostly fully represented in the classification map" is it really a limitation? Can be removed from that section.

This is added in comparison to the sentence before (incomplete representation of thin young ice areas). 'Comparatively' is now added to clarify.

L425. Convolutional neural networks also deserve being mentioned as a potential tool. Deep learning-related methods are definitely important tools for sea ice classification, but this sentence only talks about potential utilization of different forms of image texture. A sentence is added to the end of this section to mention the future use of CNN in the classification.

**Reviewer 2**

Review to submitted manuscript; "Sea ice classification of TerraSAR-X ScanSAR images for the MOSAiC expedition incorporating per-class incidence angle dependency of image texture". The manuscript investigates per-class sea ice incidence angle dependencies in TerraSAR-X ScanSAR images and GLCM textures and trains a Bayesian classifier to classify sea ice surrounding the MOSAiC expedition.

Thank you for a well written manuscript with strong English and interesting results covering a highprofile scientific campaign. To summarize the main critique points: the paper is too long, convoluted to read at times, and it is difficult to keep track of discussed subjects. In addition, parts of the methodology needs to be further clarified.

I agree with the other reviewer that the manuscript would be better suited split into two, and resubmitting them with major revisions. One manuscript could focus on the IA dependency of the TSX SC intensity and the GLCM textures, while the other could examine the GIA and the time-series of the MOSAiC campaign.

**Major Comments**

Disclaimer. I have limited practical experience with bayesian classifiers but extensive knowledge of deep learning with emphasis on sea ice segmentation using convolutional neural networks. Reviewing the methodology regarding the bayesian classifier raises the following concerns for me, which I would like you to consider and address:

• Limited testing (validation in your words) examples rectangles  $\circ$  10 reference 3x3 pixels for each class is selected for each reference scene (13 scenes in total). This is a total of 1,170 pixels for each class. Considering the abundance of data at your disposal (>1,000 x >1,000 pixels in each image?), I would refrain from needle picking select small areas. Labelling data is a time consuming task but there are tools available, which could assist, e.g. https://github.com/ESA-PhiLab/iris. At least I would require a justification for the approach.

• Small size of testing rectangles

• Why are 3 x 3 pixel rectangles selected? Could they be larger? Why not? Do the pixels have to be separate or could you label an area with multiple classes?

We aim to standardize the training/testing pixels across different classes. 3x3 pixel rectangles are selected considering typical widths of linear or small features, mainly leads, young ice areas, deformation features, and small, homogeneous ice floes. We also try to keep a relatively even distribution of polygons in each scene, thus adjacent polygons are far away from each other (roughly larger than 50 pixels), an example of which can be seen on fig.3. This approach has been used by one of our recent studies, but is not elaborated in the text of this manuscript. The above information is now added to the text to improve clarity.

**• Spatial and temporal biased training and testing**

• Generally training and testing should be carried out on areas without spatial or temporal correlation, i.e. on different scenes to avoid biases spilling over from the training to the testing phase. As the data is randomly split in training and test, I fear that some pixels may lie very close together, and could

artificially improve the model performance but without carryover to generalization of the classifier (i.e. may not be as reliable on non-testing data).

As now mentioned in the text, reference polygons are selected to be far from each other to make for a relatively even distribution over the scene. These polygons, not the pixels inside, are randomly split into training and testing. Therefore, the resulting training and testing polygons/pixels still keep a reasonable distance from each other.

More information on how the classifier is trained should be included. How is it optimized? Assuming linear IA dependence, the class-specific IA slopes and intercepts for each feature (TSX HH intensities and textures) are estimated using values from the training pixels. These IA dependence parameters are then used to calculate linearly variable mean vectors and subsequently covariance matrices which characterize the class distributions, thus fitting the classifier to the training data.

These technical details can all be found in the paper introducing the GIA classifier and is thus omitted from this manuscript given its length. A note is added to clarify this point: 'Sea ice classification of the time series is conducted using the GIA classifier trained with HH intensities and textures with optimal parameterization. **Details of the training process can be found in Lohse et al., 2021**.'

**Data**

The data selection should be more clearly explained or alternatively visualized using the acquisition dates. 53 scenes are used in this study, 50 during the MOSAiC campaign, 3 afterwards with low IAs to complete the spectrum. 40 of these scenes are not used for training the classifier (as I understand it). 13 of the 53 scenes have 10 3x3 rectangles labelled and used for training and testing.

All scenes are within the MOSAiC period. This paragraph is edited for better clarity and avoid misunderstanding: please see lines 93 - 107 in the revised manuscript.

Generally, when optimizing models, data is typically split into training, validation and testing and if supervised methodologies are applied, each split will have raw data (X) and a reference (Y), i.e. the "ground truth". Typically, a validation subset should be utilized for decision making during the optimization process, i.e. should we stop (early stopping), should we tweak the learning or regularization parameters? And finally the model performance is evaluated on the test data, which no optimization changes have been made upon. As I understand the GIA training process, you are using a test subset, and should call it as such. In regards to the segmentation tools applied, personally, I would have chosen to apply convolutional neural networks. At least it should be mentioned as a potential area of future work.

The terms 'training and testing' are now used across the text. This study has a specific aim to demonstrate class-specific IA dependencies, and thus chose a classifier which specifically incorporates this phenomenon. Yes CNN is surely a powerful tool for image classification, and a sentence is added to the end of this section to mention its potential use in the future.

**Minor Comments**

L54: Why does the TSX SC data only come in the HH polarization?

A decision was made for all TSX SC scenes for MOSAiC to be acquired in the HH polarization for consistency and to enable comparison with C-band SAR which typically come in HH+HV.

L128: 10 reference rectangles of 3 x 3 pixels sounds very small. That is only 90 pixels per class per scene, i.e. 1170 pixels. Is every class represented in every scene? And how certain are you of your qualitative selection?

See reply to the first major comment. Yes every class is represented in every scene in an equal amount. The qualitative selection is aided by co-authors who participated in the MOSAiC campaign and have extensive knowledge of the ice conditions along the expedition. A comparison between the classified results and a manual sea ice categorization map is shown in fig.8. The lack of continuous in-situ observation of sea ice type through the time series, which is the only definite 'ground truth,' is mentioned in the section 'Limitations and future steps.'

L129: Improving consistency between training scenes using a 40 km x 40 km area is unclear to me. How does this work?

This is indeed confusing to the reader, and non-essential to the work. The use of this extra 40km x 40km square cut of the scenes is now removed.

L180: Only textures of HH intensities have a consistent relationship with IA.. HH intensities as opposed to what? Or is it referring to the scaling of the image, i.e. dB.

This is now clarified as: 'In an initial examination of GLCM textures, we found that only textures of HH intensities in the logarithmic (dB) domain have a consistent linear relationship with IA, given properly constrained IA range (more details below), while textures of HH intensities in the linear domain do not.'

L337: 'On the contrary..' This sentence is quite difficult to read. I think it should be split up into two sentences.

**Edited.**

In addition, there is a pdf document attached with grammatical suggests. These have been integrated into the text.

---

## Referee Report (RR1)

Review of the second version of the submitted manuscript; *"Sea ice classification of TerraSAR-X ScanSAR images for the MOSAiC expedition incorporating per-class incidence angle dependency of image texture"*. The authors have made comprehensive revisions to the manuscript and included many of the previous suggestions from both reviewers as well as elaborated on key points requested in the previous manuscript version increasing the readability.

**General Comments**

The paper has been shortened by two pages, however, I must admit that I would still prefer if the manuscript could be trimmed further. For instance, Figure 2 and the associated text in L117-131 serve the purpose to argue for the need to split young ice into multiple classes. However, the story regarding high winds that may have caused an opening in the sea ice, I see as less relevant to this narrative.

I am happy that several of my concerns regarding the methodology are addressed in the revision, particularly increase in reference polygons from 10 to 15 per scene as well as ensuring that the spatial separation for testing was taken into account to give a more generalistic evaluation of the classifier. In addition, the inclusion of previous work and referencing highlighting the classifier as a established method is appreciated. Though personally, I would have preferred to label a much larger number of pixels but I will accept the approach in its given form.

There are still a few things, which I would like to have elaborated as well as a number of minor adjustments, e.g. grammatical suggestions and references in need of edits.

**Major Comments**

L13, there was a change in accuracy from 86.05% to 83.70%. Was this a result of the increase of the number of reference polygons? (L19-20 in the changes document).

Figure 7. Is this showing the distribution of classification accuracy for the different classes or the accuracy distribution for the test scenes? In case of the latter, what is the average accuracy based on, all pixels or average scene accuracy?
In addition, I think you should consider investigating the raw numerical performance of your classifier more. For instance, you could examine what incorrect classifications are labelled as. Are there patterns?

In relation to the beginning of the 'Results and Discussion' section on page 13, I think it would be beneficial to the reader if a short summary of what you go through, e.g. "first we investigate the performance of the classifier.. Then we compare with sea ice roughness estimates.. After which we present the time-series of ice class fractions.. And then we present the limitations and future steps...

**Minor Comments**

Table 2. The GLCM textures are defined but variables are not described. I think you should consider adding the names of variables for instance in the table caption. E.g. i, j, $\mu$, P, HXY, etc..

L373: You have not explained what the abbreviation of CNN is.There are also some works that use CNNs to classify sea ice types. Perhaps it could be useful for you to investigate. E.g. Boulze, H., Korosov, A. & Brajard, J. Classification of Sea Ice Types in Sentinel-1 SAR Data Using Convolutional Neural Networks. *Remote Sensing* **12**, 2165 (2020).
(though there are probably more recent developments)

Additional suggestions for grammatical adjustments can be found in the attached PDF.

[revised manuscript text omitted]

| | |
|---|---|
| (1) Cluster Prominence (CLP): $\sum_i \sum_j \left(i+j-\mu_i-\mu_j\right)^4 P_{i,j}$ | (10) Homogeneity (HOM): $\sum_i \sum_j \frac{P_{i,j}}{1+(i-j)^2}$ |
| (2) Cluster Shade (CLS): $\sum_i \sum_j \left(i+j-\mu_i-\mu_j\right)^3 P_{i,j}$ | (11) Information Measure of Correlation 1 (IMC1): $\frac{HXY-HXY1}{max(HX,HY)}$ |
| (3) Contrast (CON): $\sum_i \sum_j P_{i,j}\left(i-j\right)^2$ | (12) Information Measure of Correlation 2 (IMC2): $\sqrt{\left(1-exp\left(-2.0\left(HXY2-HXY\right)\right)\right)}$ |
| (4) Correlation (COR): $\frac{\sum_i \sum_j ijP_{i,j}-\mu_x\mu_y}{\sigma_x\sigma_y}$ | (13) Maximum Probability (MXP): $max\left(P_{i,j}\right)$ |
| (5) Difference Entropy (DFE): $-\sum_{i=0}^{N_g-1} P_{x-y}\left(i\right)logP_{x-y}\left(i\right)$ | (14) Mean (MEAN): $\sum_i \sum_j iP_{i,j}$ |
| (6) Difference Variance (DFV): $\sum_{i=2}^{2N_g}\left(i-\left[\sum_{i=2}^{2N_g}iP_{x-y}\left(i\right)\right]\right)^2$ | (15) Sum Average (SMA): $\sum_{i=2}^{2N_g}iP_{x+y}\left(i\
[revised manuscript text omitted]

---

## Author Response (AR2)

**Reviewer 1**

The clarity and logics of the manuscript has significantly improved during the revision. I have only few minor comments and can recommend the manuscript for publication after the corrections.

L210. Detailed steps ... are ...

Edited.

L304 - 316. This paragraph became quite controversial. The logarithmic fit can indeed be seen, but as you write below, the spread of the scatter plot is quite wide. I don't think it is correct to attribute this spread only to the difference between surface/bottom roughness seen by GEM-2 and the roughness articulated on the SAR image. The spread can also appear because the HH intensities (especially on X-band SAR) are controlled NOT only by geometric ice roughness. Therefore, separating FYI and MYI into different degrees of deformation cannot be justified globally. I'm afraid, that the results obtained in a relatively limited dataset can be misleading and should be presented with caution. I would suggest the following phrasing:

This indicates that TSX SC HH intensities in these particular transects are controlled largely by geometric ice roughness because other sea ice surface properties (micro-roughness, salinity, snow, etc.) are quite similar.

This relationship, ..., illustrates that under similar environmental conditions FYI and MYI can be attributed to different degrees of deformation.

Edited.

Figure 11. Y-labels for the upper part of the plot of 'Class fractions' are missing.

Edited.

**Reviewer 2**

Review of the second version of the submitted manuscript; "Sea ice classification of *TerraSAR-X ScanSAR images for the MOSAiC expedition incorporating per-class incidence angle dependency of image texture*". The authors have made comprehensive revisions to the manuscript and included many of the previous suggestions from both reviewers as well as elaborated on key points requested in the previous manuscript version increasing the readability.

**General Comments**

The paper has been shortened by two pages, however, I must admit that I would still prefer if the manuscript could be trimmed further. For instance, Figure 2 and the associated text in L117-131 serve the purpose to argue for the need to split young ice into multiple classes.

However, the story regarding high winds that may have caused an opening in the sea ice, I see as less relevant to this narrative.

The sentence regarding the wind effect on sea ice breakup has been deleted, and this paragraph has been edited for better clearer and more concise, as follows. The whole text has been read through and edited again for conciseness.

Young ice shows a wide range of HH intensities due to differences in surface characteristics, which affects ice type classification. Fig. 2 shows an example of the progression of young ice on overlapping TSX SC and S1 EW scenes , both in HH . High winds were observed during this period (Krumpen et al., 2021), which presumably contributed to the ice opening and deformation events. in HH polarization. On 2019.11.20, wide-spread lead openings occurred around the CO. Between 2019.11.20 and 2019.11.21, more openings appeared which quickly re-froze into young ice. For On the TSX scenes, on 2019.11.21, most of the young ice areas appear very bright. While the leads gradually close upSubsequently, young ice gradually darken to similar brightness to the surrounding ice. On the S1 EW-scenes, HH intensities for of young ice gradually increase, from similar or lower brightness to nearby level ice to very bright on 2019.11.23 and 2019.11.24, until they again reach. Afterwards, they again darken to similar brightness to the surroundings to the surroundingslater. The changing young ice intensities through time is are presumably due to evolving surface roughness, e.g., influenced by the growing and disappearing formation and eyolution of frost flowers which are highly saline with varying sizes, and eauses changing leading to varying scales of surface

6

roughness which strongly influences SAR signals (Martin et al., 1995; Barber et al., 2014; Isleifson et al., 2018; Johansson et al., 2018). The delayed increase and decrease in young ice intensities backscatter in C-band (5.405 GHz) compared to X-band (9.65 GHz) data is then presumably due to different interactions between changing surface roughness scales and different SAR wavelengths (Isleifson et al., 2010; Dierking, 2010; Barber et al., 2014; Park et al., 2020). These observations confirm the need to split young ice into separate classes for ice type classification, which is described below.

I am happy that several of my concerns regarding the methodology are addressed in the revision, particularly increase in reference polygons from 10 to 15 per scene as well as ensuring that the spatial separation for testing was taken into account to give a more generalistic evaluation of the classifier. In addition, the inclusion of previous work and referencing highlighting the classifier as a established method is appreciated. Though personally, I would have preferred to label a much larger number of pixels but I will accept the approach in its given form.

There are still a few things, which I would like to have elaborated as well as a number of minor adjustments, e.g. grammatical suggestions and references in need of edits.

**Major Comments**

L13, there was a change in accuracy from 86.05% to 83.70%. Was this a result of the increase of the number of reference polygons? (L19-20 in the changes document).

This is a result of the overall change of workflow as suggested by both reviewers, i.e., changed number of reference polygons, the removal of speckle filtering, and the resulting change in the chosen combination of GLCM textures to use. Qualitatively this change of accuracy does not significantly change the conclusions

of this work or the quality of the product, but the workflow is now more theoretically sound.

Figure 7. Is this showing the distribution of classification accuracy for the different classes or the accuracy distribution for the test scenes? In case of the latter, what is the average accuracy based on, all pixels or average scene accuracy?

In addition, I think you should consider investigating the raw numerical performance of your classifier more. For instance, you could examine what incorrect classifications are labelled as. Are there patterns?

The latter, based on all the testing pixels of each scene.

An explanation of the misclassification patterns is now added to the text: 'For the final classification with MRF contextual smoothing, the confusion matrix (not shown) indicates that remaining misclassifications mostly happen between the difficult class pairs, as expected. Leads and level ice are mostly correctly classified.' We feel this clarification is sufficient to elaborate the classification performance without adding too much to the text.

In relation to the beginning of the 'Results and Discussion' section on page 13, I think it would be beneficial to the reader if a short summary of what you go through, e.g. "first we investigate the performance of the classifier.. Then we compare with sea ice roughness estimates.. After which we present the time-series of ice class fractions.. And then we present the limitations and future steps...

This is added to the text: 'In this section, we first present qualitative and quantitative evaluation of the performance of our classification product. Then, we compare the classification maps with sea ice roughness estimates from MOSAiC in-situ data, and accordingly evaluate our classification scheme splitting FYI and MYI into different deformation states. To evaluate the consistency of the classification, temporal development of areal fractions of each class is then presented and compared with indicators of ice openings from in-situ data and other MOSAiC studies. Finally, we list several limitations of our workflow and give potential directions for future studies following this work.'

Sentences with similar information in the preceding paragraph has accordingly been deleted.

**Minor Comments**

Table 2. The GLCM textures are defined but variables are not described. I think you should consider adding the names of variables for instance in the table caption. E.g. i, j,  $\mu$ , P, HXY, etc..

**Added to the text.**

L373: You have not explained what the abbreviation of CNN is. There are also some works that use CNNs to classify sea ice types. Perhaps it could be useful for you to investigate.

E.g. Boulze, H., Korosov, A. & Brajard, J. Classification of Sea Ice Types in Sentinel-1 SAR Data Using Convolutional Neural Networks. *Remote Sensing* **12**, 2165 (2020).

(though there are probably more recent developments)

Full name and citation added.

Additional suggestions for grammatical adjustments can be found in the attached PDF.

Edited according to the comments.

**References:**

L422:: the DOI link does not work

Edited.

L 427: the link contains two https://doi.org.. This should be fixed.

**Edited.**

L32: the DOI link does not work, though I believe it is due to the .ch13. Perhaps this should be written instead of included in the DOI?

**Edited.**

L444: please fix the DOI link

**Edited.**

L454: the scihub link does not work

**Edited.**

L560: the link contains two https://doi.org.. This should be fixed.

**Edited.**